# UNIVERSAL SPEECH ENHANCEMENT WITH SCORE-BASED DIFFUSION

## ABSTRACT

Removing background noise from speech audio has been the subject of considerable effort, especially in recent years due to the rise of virtual communication and amateur recordings. Yet background noise is not the only unpleasant disturbance that can prevent intelligibility: reverb, clipping, codec artifacts, problematic equalization, limited bandwidth, or inconsistent loudness are equally disturbing and ubiquitous. In this work, we propose to consider the task of speech enhancement as a holistic endeavor, and present a universal speech enhancement system that tackles 55 different distortions at the same time. Our approach consists of a generative model that employs score-based diffusion, together with a multi-resolution conditioning network that performs enhancement with mixture density networks. We show that this approach significantly outperforms the state of the art in a subjective test performed by expert listeners. We also show that it achieves competitive objective scores with just 4–8 diffusion steps, despite not considering any particular strategy for fast sampling. We hope that both our methodology and technical contributions encourage researchers and practitioners to adopt a universal approach to speech enhancement, possibly framing it as a generative task.

## 1 INTRODUCTION

Real-world recorded speech almost inevitably contains background noise, which can be unpleasant and prevent intelligibility. Removing background noise has traditionally been the objective of speech enhancement algorithms (Loizou, 2013). Since the 1940s (Kolmogorov, 1941; Wiener, 1949), a myriad of denoising approaches based on filtering have been proposed, with a focus on stationary noises. With the advent of deep learning, the task has been dominated by neural networks, often outperforming more classical algorithms and generalizing to multiple noise types (Lu et al., 2013; Pascual et al., 2017; Rethage et al., 2018; Défossez et al., 2020; Fu et al., 2021). Besides recent progress, speech denoising still presents room for improvement, especially when dealing with distribution shift or real-world recordings.

Noise however is only one of the many potential disturbances that can be present in speech recordings. If recordings are performed in a closed room, reverberation is ubiquitous. With this in mind, a number of works have recently started to zoom out the focus in order to embrace more realistic situations and tackle noise and reverberation at the same time (Su et al., 2021; 2019; Polyak et al., 2021). Some of these works adopt a generation or re-generation strategy (Maiti & Mandel, 2019), in which a two-stage approach is employed to first enhance and then synthesize speech signals. Despite the relative success of this strategy, it is still an open question whether such approaches can perceptually outperform the purely supervised ones, especially in terms of realism and lack of voice artifacts.

Besides noise and reverberation, a few works propose to go one step further by considering additional distortions. Pascual et al. (2019) introduce a broader notion of speech enhancement by considering whispered speech, bandwidth reduction, silent gaps, and clipping. More recently, Nair & Koishida (2021) consider silent gaps, clipping, and codec artifacts, and Zhang et al. (2021a) consider clipping and codec artifacts. In concurrent work, Liu et al. (2021) deal with bandwidth reduction and clipping in addition to noise and reverberation. Despite the recent efforts to go beyond pure denoising, we are not aware of any speech enhancement system that tackles more than 2–4 distortions at the same time.

In this work, we take a holistic approach and regard the task of speech enhancement as a universal endeavor. We believe that, for realistic speech enhancement situations, algorithms need not only face

and improve upon background noise and possibly reverberation, but also to correct a large number of typical but usually neglected distortions that are present in everyday recordings or amateur-produced audio, such as bandwidth reduction, clipping, codec artifacts, silent gaps, excessive dynamics compression/expansion, sub-optimal equalization, noise gating, and others (in total, we deal with 55 distortions, which can be grouped into 10 different families).

Our solution relies on an end-to-end approach, in which a generator network synthesizes clean speech and a conditioner network informs of what to generate. The idea is that the generator learns from clean speech and both generator and conditioner have the capability of enhancing representations, with the latter undertaking the core part of this task. For the generator, we put together a number of known and less known advances in score-based diffusion models (Sohl-Dickstein et al., 2015; Song & Ermon, 2019; Ho et al., 2020). For the conditioner, we develop a number of improved architectural choices, and further propose the usage of auxiliary, out-of-path mixture density networks for enhancement in both the feature and the waveform domains. We quantify the relative importance of these main development steps using objective metrics, and show how the final solution outperforms the state of the art in all considered distortions using a subjective test with expert listeners (objective metrics for the denoising task are also reported in the Appendix). Finally, we also study the number of diffusion steps needed for performing high-quality universal speech enhancement, and find it to be on par with the fastest diffusion-based neural vocoders without the need for any specific tuning.

## 2 RELATED WORK

Our approach is based on diffusion models (Sohl-Dickstein et al., 2015; Song & Ermon, 2019; Ho et al., 2020). While diffusion models have been more extensively studied for unconditional or weakly-conditioned image generation, our work presents a number of techniques for strongly-conditioned speech re-generation or enhancement. Diffusion-based models achieve state-of-the-art quality on multiple generative tasks, in different domains. In the audio domain, they have been particularly successful in speech synthesis (Chen et al., 2021; Kong et al., 2021), text-to-speech (Jeong et al., 2021; Popov et al., 2021), bandwidth extension (Lee & Han, 2021), or drum sound synthesis (Rouard & Hadjeres, 2021). An introduction to diffusion models is given in Appendix A

Diffusion-based models have also recently been used for speech denoising. Zhang et al. (2021a) expand the DiffWave vocoder (Kong et al., 2021) with a convolutional conditioner, and train that separately with an L1 loss for matching latent representations. Lu et al. (2021) study the potential of DiffWave with noisy mel band inputs for speech denoising and, later, Lu et al. (2022) and Welker et al. (2022) propose formulations of the diffusion process that can adapt to (non-Gaussian) real audio noises. These studies with speech denoising show improvement over the considered baselines, but do not reach the objective scores achieved by state-of-the-art approaches (see also Appendix E). Our work stems from the WaveGrad architecture (Chen et al., 2021), introduces a number of crucial modifications and additional concepts, and pioneers universal enhancement by tackling an unprecedented amount of distortions.

The state of the art for speech denoising and dereverberation is dominated by regression and adversarial approaches (Défossez et al., 2020; Fu et al., 2021; Su et al., 2021; Isik et al., 2020; Hao et al., 2021; Kim & Seo, 2021; Zheng et al., 2021; Kataria et al., 2021). However, if one considers further degradations of the signal like clipping, bandwidth removal, or silent gaps, it is intuitive to think that generative approaches have great potential (Polyak et al., 2021; Pascual et al., 2019; Zhang et al., 2021b), as such degradations require generating signal where, simply, there is none. Yet, to the best of our knowledge, this intuition has not been convincingly demonstrated through subjective tests involving human judgment. Our work sets a milestone in showing that a generative approach can outperform existing supervised and adversarial approaches when evaluated by expert listeners.

## 3 DIFFUSION-BASED UNIVERSAL SPEECH ENHANCEMENT

### 3.1 METHODOLOGY

**Data —** To train our model, we use a data set of clean and programmatically-distorted pairs of speech recordings. To obtain the clean speech, we sample 1,500 h of audio from an internal pool of data sets and convert it to 16 kHz mono. The speech sample consists of about 1.2 M utterances

of between 3.5 and 5.5 s, from thousands of speakers, in more than 10 languages, and with over 50 different recording conditions (clean speech is chosen to be the most dry possible and of high quality). To validate our model, we use 1.5 h of clean utterances sampled from VCTK (Yamagishi et al., 2019) and Harvard sentences (Henter et al., 2014), together with noises/backgrounds from DEMAND (Thiemann et al., 2013) and FSDnoisy18k (Fonseca et al., 2019). Train data does not overlap with the validation partition nor with other data used for evaluation or subjective testing.

To programmatically generate distorted speech, we consider 10 distortion families: band limiting, codec artifacts, signal distortion, loudness dynamics, equalization, recorded noise, reverb, spectral manipulation, synthetic noise, and transmission. Each family includes a variety of distortion algorithms, which we generically call 'types'. For instance, types of synthetic noise include colored noise, electricity tones, non-stationary noise bursts, etc., types of codecs include OPUS, Vorbis, MP3, EAC3, etc., types of reverb include algorithmic reverb, delays, and both real and simulated room impulse responses, and so on. In total, we leverage 55 distortion types. Distortion type parameters such as strength, frequency, or gain are set randomly within reasonable bounds. A more comprehensive list of distortion families, types, and parameters can be found in Appendix C.

**Evaluation —** To measure relative improvement, we use objective speech quality metrics reflecting different criteria. On the one hand, we employ speech enhancement metrics COVL (Loizou, 2013) and STOI (Taal et al., 2010), which are widely used for denoising or dereverberation tasks. On the other hand, we employ the codec quality metric WARP-Q (Jassim et al., 2021) and an internal semi-supervised metric imitating SESQA (Serrà et al., 2021b), which should perform well with generative algorithms with perceptually-valid outputs that do not necessarily perfectly align with the target signal. We also consider the composite measure COMP that results from normalizing the previous four metrics between 0 and 10 and taking the average.

To compare with the state of the art, we perform a subjective preference test. In it, listeners are presented with a reference distorted signal plus two enhanced versions of it: one by an existing approach and one by the proposed approach. Then, they are asked to choose which of the two enhanced signals they prefer, based on the presence of the original nuisance, voice distortion, audible artifacts, etc. The test was voluntarily performed by 22 expert listeners, each of them listening to randomly-chosen pairs of distorted and enhanced signals, taken from the online demo/example pages of 12 existing approaches, plus the corresponding version enhanced by our approach. Further details of our evaluation methodology are provided in Appendix D.

## 3.2 BASE APPROACH

**Score-based diffusion —** In this work, we use a variance exploding (VE) diffusion approach (Song et al., 2021). We train our score network $S$ following a denoising score matching paradigm (Song & Ermon, 2019; 2020), using

$$\mathcal{L}_{\text{SCORE}} = \mathbb{E}_t \mathbb{E}_{\mathbf{z}_t} \mathbb{E}_{\mathbf{x}_0} \left[ \frac{1}{2} \left\| \sigma_t S(\mathbf{x}_0 + \sigma_t \mathbf{z}_t, \mathbf{c}, \sigma_t) + \mathbf{z}_t \right\|_2^2 \right], \tag{1}$$

where $t \sim \mathcal{U}(0,1)$, $\mathbf{z}_t \sim \mathcal{N}(\mathbf{0}, \mathbf{I})$, $\mathbf{x}_0 \sim p_{\text{data}}$, $\mathbf{c}$ is the conditioning signal, and values $\sigma_t$ follow a geometric noise schedule (Song & Ermon, 2019). In all our experiments we use $\sigma_0 = 5 \cdot 10^{-4}$ and $\sigma_1 = 5$, which we find sufficient for audio signals between $-1$ and $1$ (cf. Song & Ermon, 2020). We consider different approaches to obtain $\mathbf{c}$, but all of them have $\tilde{\mathbf{x}}_0$, the distorted version of $\mathbf{x}_0$, as the main and only input ($\mathbf{x}_0$ and $\tilde{\mathbf{x}}_0$ are both in the waveform domain).

To sample, we follow noise-consistent Langevin dynamics (Jolicoeur-Martineau et al., 2021), which corresponds to the recursion

$$\mathbf{x}_{t_{n-1}} = \mathbf{x}_{t_n} + \eta \sigma_{t_n}^2 S(\mathbf{x}_{t_n}, \mathbf{c}, \sigma_{t_n}) + \beta \sigma_{t_{n-1}} \mathbf{z}_{t_{n-1}}$$

over $N$ uniformly discretized time steps $t_n \in [0,1]$, where we set $\eta$ and $\beta$ with the help of a hyperparameter $\epsilon \in [1, \infty)$. An extensive derivation of our approach to both training and sampling can be found in Appendix A, including an introduction to score-based diffusion.

**General model description —** We use convolutional blocks and a couple of bi-directional recurrent layers. Convolutional blocks are formed by three 1D convolutional layers, each one preceded by a multi-parametric ReLU (PReLU) activation, and all of them under a residual connection. If needed,

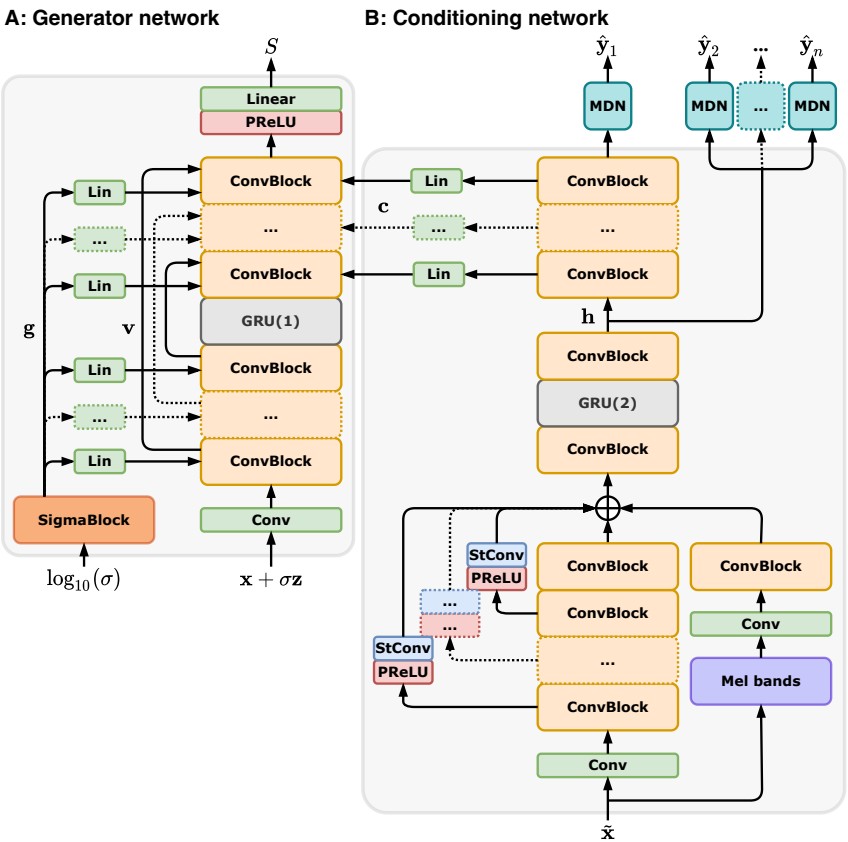

Figure 1: Block diagram of the proposed approach. Individual blocks are depicted in Appendix B.

up- or down-sampling is applied before or after the residual link, respectively (Appendix B). We perform up-/down-sampling with transposed/strided convolutions, halving/doubling the number of channels at every step. The down-sampling factors are {2,4,4,5}, which yield a 100 Hz latent representation for a 16 kHz input.

The model consists of a generator and a conditioning network (Fig. 1). The generator network is formed by a UNet-like structure with skip connections $\mathbf{v}$ and a gated recurrent unit (GRU) in the middle (Fig. 1-A). Convolutional blocks in the generator receive adaptor signals $\mathbf{g}$, which inform the network about the noise level $\sigma$, and conditioning signals $\mathbf{c}$, which provide the necessary speech cues for synthesis. Signals $\mathbf{g}$ and $\mathbf{c}$ are mixed with the UNet activations using FiLM (Perez et al., 2018) and summation, respectively. To obtain $\mathbf{g}$, we process the logarithm of $\sigma$ with random Fourier feature embeddings and an MLP as in Rouard & Hadjeres (2021) (see also Appendix B). The conditioning network processes the distorted signal $\tilde{\mathbf{x}}$ with convolutional blocks featuring skip connections to a down-sampled latent that further exploits log-mel features extracted from $\tilde{\mathbf{x}}$ (Fig. 1-B). The middle and decoding parts of the network are formed by a two-layer GRU and multiple convolutional blocks, with the decoder up-sampling the latent representation to provide multi-resolution conditionings $\mathbf{c}$ to the generator. Multiple heads and target information are exploited to improve the latent representation and provide a better $\mathbf{c}$ (see below). We call our approach UNIVERSE, for universal speech enhancer.

## 3.3 DEVELOPING UNIVERSE

In the following, we explain all the steps we took to arrive at the above-mentioned structure, motivating each decision and quantifying its impact with objective metrics (schematic diagrams are depicted in Fig. 2). Unless stated otherwise, all models are trained with Adam and weight decay for 1 M iterations, with two-second long audio frames and a batch size of 32. We use a cosine learning rate schedule with a linear warm-up that spans the first 5 % of the iterations. Under these specifications,

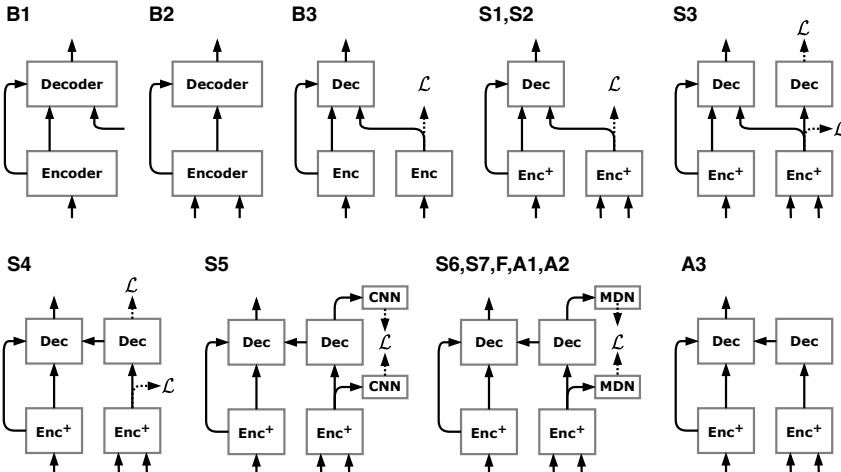

Figure 2: Diagrams of the followed steps: $\mathcal{L}$ indicates auxiliary losses and $^+$ capacity increase.

training a UNIVERSE model with 49 M parameters takes less than 5 days using two Tesla V100 GPUs and PyTorch's native automatic mixed precision (Paszke et al., 2019).

**Initial baselines (B1–B3)** — We start with a structure inspired by WaveGrad (Chen et al., 2021), featuring a UNet-like architecture with the aforementioned convolutional blocks, each with 64, 128, 256, and 512 channels (Sec. 3.2 and Appendix B). To condition we use the distorted speech $\tilde{\mathbf{x}}$, from which we compute a 100 Hz log-mel representation with 80 bands that is summed to the UNet latent after a linear projection (Fig. 2-B1). This approach turns out to lack enhancement capabilities, probably hampered by the coarse mel representation of $\tilde{\mathbf{x}}$. It also presents considerable speech distortion, probably due to the distortions in the mel representation (Table 1, B1).

We next consider an approach inspired by NU-Wave (Lee & Han, 2021). We employ the same convolutional UNet structure as before, but inject the conditioning signal $\tilde{\mathbf{x}}$ at the input (Fig. 2-B2), concatenated with $\mathbf{x} + \sigma\mathbf{z}$ (see Eq. 1). We find that this approach yields less speech distortion and better scores than the previous one (Table 1, B2). However, after listening to some excepts, we notice that a lot of the audio is 'bypassed' from input to output. This leads to, for example, input noises present at the output, or silent gaps not being reconstructed. We hypothesize that the UNet's skip connections are the culprit for this behavior (and removing them affected training in a dramatic way).

Another baseline approach we consider is inspired by ModDW (Zhang et al., 2021a). In it, following DiffWave (Kong et al., 2021), an auxiliary encoder and loss are used to learn a conditioning latent (Fig. 2-B3). This encoder uses the same blocks as the generative one, but with progressively down-sampled skip connections. Also different from ModDW, we use mean squared error (MSE) instead of an L1 loss, and compare directly with the clean mel-band representation instead of a pre-learnt latent. With these two modifications, we find no problems in training the model end-to-end, in contrast to the original ModDW, which had to be trained using separate stages (Zhang et al., 2021a). The scores for this approach are in the middle of the previous baselines, closer to the second one (Table 1, B3).

**Capacity and losses (S1–S7)** — With the three baselines above, we decide to add processing capacity in order to obtain a better conditioning signal. This decision was motivated by preliminary analysis showing that B1 and B3 were capable to synthesize high-quality speech in a vocoder setting (that is, when a clean mel-band conditioning was provided). Hence, with more capacity in the conditioning network, one should come closer to such clean mel-band scenario (an observation also made by Zhang et al. (2021a)). Unfortunately, increasing the capacity for the best scoring model, B2, could not avoid the bypass problems outlined above. Therefore, we focus on improving B3, which had the best scores after B2 and a similar listening quality, with less noise but more speech distortion.

We first add an extra log-mel input to the conditioner's encoder, with a convolutional block after it, and sum its output to the skip connections coming from the distorted waveform processing (mels are extracted from the same distorted signal $\tilde{\mathbf{x}}$ that is input to the waveform encoder, see Fig. 1-B). This

Table 1: Test set objective scores for the steps in developing UNIVERSE. Increments $\Delta$ are calculated with respect to the COMP value of the previous row, except for A1–A3, which take S7 as reference.

| ID | Description | COVL ↑ | STOI ↑ | WARP-Q ↓ | SESQA ↑ | COMP ↑ | $\Delta$ ↑ |
|----|-------------|--------|--------|----------|---------|--------|-----------|
| B1 | Inspired by WaveGrad | 1.55 | 0.755 | 0.952 | 4.90 | 3.77 | |
| B2 | Inspired by NU-Wave | 2.02 | 0.829 | 0.873 | 5.57 | 4.69 | |
| B3 | Inspired by ModDW | 1.76 | 0.811 | 0.902 | 5.35 | 4.34 | |
| S1 | Extra mel input | 1.86 | 0.820 | 0.872 | 5.51 | 4.54 | +0.20 |
| S2 | Latent RNN+CNN | 1.99 | 0.844 | 0.834 | 5.41 | 4.75 | +0.21 |
| S3 | Auxiliary decoder & loss | 2.37 | 0.880 | 0.830 | 5.81 | 5.25 | +0.50 |
| S4 | Multi-resolution cond. | 2.53 | 0.888 | 0.811 | 6.18 | 5.54 | +0.29 |
| S5 | Out-of-path losses | 2.58 | 0.893 | 0.787 | 6.29 | 5.66 | +0.12 |
| S6 | Mixture density | 2.75 | 0.909 | 0.748 | 6.47 | 5.95 | +0.29 |
| S7 | Extra latent targets | 2.73 | 0.909 | 0.751 | 6.52 | 5.96 | +0.01 |
| F | Scaling parameters & iterations | 3.12 | 0.930 | 0.679 | 6.82 | 6.50 | +0.54 |
| A1 | Two-stage training | 2.63 | 0.904 | 0.783 | 6.35 | 5.76 | −0.20 |
| A2 | No SigmaBlock | 2.53 | 0.897 | 0.816 | 6.42 | 5.64 | −0.32 |
| A3 | No auxiliary losses | 2.55 | 0.872 | 0.767 | 6.27 | 5.59 | −0.37 |

already results in an improvement with respect to B3, with objective scores that are very close to B2 (Table 1, S1). Next, we add a two-layer GRU and two convolutional blocks after the skip summation. We also add a one-layer GRU to the generator latent. Both GRUs should provide the model with a larger receptive field and better sequential processing capabilities. With this additions, the model increases the COMP score to 4.75 (Table 1, S2).

After improving the encoders' capacity, we decide to add a waveform loss to the existing MSE on log-mel latents. To do so, we also need to include additional decoder blocks for up-sampling the latent (Fig. 2-S3). For the waveform loss, we use MSE together with a multi-resolution STFT loss (Yamamoto et al., 2020). The result is a noticeable improvement (Table 1, S3). Although this is the largest improvement in objective scores, we do not observe such a large difference throughout informal listening. Better low-level details are present, but we suspect a large part of the score improvement is due to the objective measures paying too much attention to (sometimes irrelevant) low-level detail, which is now induced by the losses in the waveform domain.

Now that we have parallel up-sampling blocks in the generator and conditioning decoders, we can condition at multiple resolutions between $100\,\mathrm{Hz}$ and $16\,\mathrm{kHz}$ (Fig. 2-S4). This provides an improvement (Table 1, S4). However, after careful listening, we have the impression that loss errors have a strong effect on the conditioning, provoking alterations that difficult the task of the generator (for instance, we hear a bit of muffled speech, presumably resulting from using MSE in both mel-band and waveform domains). To try to alleviate these issues, we decouple the loss calculations from the main signal path of the conditioning network. To do so, we use two convolutional heads (Fig. 2-S5), which include layer normalization, PReLUs, and a convolution with a kernel width of 3. With that, we observe an improvement not only in objective metrics (Table 1, S5), but also in subjective quality.

Decoupled auxiliary heads allow us to further think of alternative loss functions that might work better for latent and waveform representations. Interestingly, such loss functions do not need to be restricted to regression or adversarial approaches, as used in the literature, but can be probabilistic and model the representations' distribution explicitly. Now that losses are out of the signal path, nothing prevents us from using approaches that otherwise would require to sample from a probabilistic prediction in order to proceed with the signal flow. With the idea of better modeling the distribution of both log-mels and waveforms, we employ a mixture density network (MDN) approach (Bishop, 1994), with the same head architecture as before, but with 3 Gaussian components, each calculated from different convolutional layers (Fig. 2-S6). Assuming independent time steps, the negative log-likelihood of each time step using $k$ components of dimensionality $d$ and diagonal covariance is

$$\mathcal{L}_{\mathrm{MDN}} = -\ln\left[\sum_{i=1}^{k} \frac{\alpha^{(i)}}{(2\pi)^{d/2}\prod_{j=1}^{d} s_j^{(i)}} \exp\left\{-\frac{1}{2}\sum_{j=1}^{d}\left(\frac{y_j - m_j^{(i)}}{s_j^{(i)}}\right)^2\right\}\right],$$

where $\mathbf{y}$ is the target representation and $\alpha^{(i)}$, $\mathbf{m}^{(i)}$, and $\mathbf{s}^{(i)}$ denote the output of the convolutional layer corresponding to the mixing probability, the mean, and the standard deviation of the $i$-th Gaussian, respectively. Replacing standard losses by MDNs has a positive effect on the objective scores (Table 1, S6). Together with moving the loss computation out of the signal path (S5), they provide a COMP increment of 0.41, the largest one besides using an auxiliary decoder and loss (S3).

After introducing the MDNs, our last step consists of including more targets in the latent predictions. These additional targets are pitch and harmonicity, as provided by crepe (Kim et al., 2018), voice activity detection (VAD) and loudness, provided by simple internal algorithms, and the deltas of all of them. We consider a separate MDN for each target type. That is, one for mel bands and their deltas, one for pitch/harmonicity and their deltas, and one for VAD/loudness and their deltas. The result only provides a marginal improvement in objective scores (Table 1, S7), but we find such improvement to be consistent after listening to a number of enhanced signals.

**Scaling up (F)** — Finally, after defining our base model, we can train a larger model for a longer time. We increase the model size by doubling the number of channels and we reduce the learning rate by two. The resulting model has 189 M parameters, and is trained for 2.5 M iterations using a batch size of 64. This takes less than 14 days using 8 Tesla V100 GPUs. The result of scaling up parameters and training is a large improvement, both objectively and subjectively (Table 1, F). This is the model we will use in our final evaluation, with 50 diffusion steps (see Sec. 4.3).

**Further ablations (A1–A3)** — Starting with S7, we can further assess the effect of some design choices. For instance, it is common in speech enhancement to train multi-part models using multiple stages, or taking some pre-trained (frozen) parts (Polyak et al., 2021; Maiti & Mandel, 2019; Nair & Koishida, 2021; Liu et al., 2021). The equivalent of this strategy for our two-part model would be to first train the conditioner using all $\mathcal{L}_{\text{MDN}}$ losses, freeze it, and then train the generator with $\mathcal{L}_{\text{SCORE}}$. An intuition for this could be that, this way, the generator network is 'decoupled' from the enhancement/conditioner one, and that therefore the two networks can fully concentrate in their commitment (generating and cleaning, respectively). Nonetheless, this intuition seems to be at least partially wrong, as we obtain worse scores (Table 1, A1). In fact, this result points towards a 'coupling' situation, in which part of the generator performs some enhancement and part of the conditioner shapes the generator input.

Another design choice we can question is the strategy to provide the diffusion model with the noise level $\sigma$ (SigmaBlock, Fig. 1). This strategy is used by some audio generation models (like CRASH by Rouard & Hadjeres (2021), from which we borrow it), but other models employ other strategies or none. We observe that the use of this strategy is important for our generator network (Table 1, A2). Finally, we can also quantify the effect of the additional losses $\mathcal{L}_{\text{MDN}}$. By removing them and training the whole model only with $\mathcal{L}_{\text{SCORE}}$, we observe a clear decrease in objective scores (Table 1, A3). Thus, we conclude that both auxiliary noise levels and losses are important.

# 4 RESULTS

## 4.1 COMPARISON WITH EXISTING APPROACHES

To compare with the state of the art, a common approach is to use objective metrics and well-established test sets. However, since the task of universal speech enhancement has not been formally addressed before, an established test set does not exist. Furthermore, it is not yet clear if common objective metrics provide a reasonable measurement for the enhancement of other distortions beyond additive noise and reverberation. An alternative is to evaluate on separate, individual test sets, each of them focused on a particular task (for example denoising, declipping, codec artifact removal, and so on). However, to the best of our knowledge, there do not exist well-established test sets nor metrics for other tasks beyond denoising. Therefore, to be the most fair possible to existing approaches, and in order to skip potentially flawed objective metrics, we think the best approach is to conduct a subjective test with expert listeners (Sec. 3.1 and Appendix D). Nonetheless, in Appendix E we also report objective scores for the two most common denoising test sets and show that the proposed approach achieves competitive results despite being generative and not favored by objective metrics.

In our subjective test, we compare against 12 existing approaches on different combinations of enhancement tasks (Table 2). We find that UNIVERSE outperforms all existing approaches by a

Table 2: Preference test results, including an indication of the model class (regression, adversarial, generative) and the considered distortions (noise, reverb, bandwidth reduction, clipping, codec artifacts, and others). Subjects' preference (other/existing approach or UNIVERSE) is shown on the right, together with statistical significance $\star$ (binomial test, $p < 0.05$, Holm-Bonferroni adjustment).

| Approach | Class | Distortions | | | | | | Preference (%) | |
|---|---|---|---|---|---|---|---|---|---|
| | | Noise | Rev | BW | Clip | Codec | Others | Other | UNIVERSE |
| Demucs (Défossez et al., 2020) | Reg | ✓ | | | | | | 2.3 | **97.7** $\star$ |
| MetricGAN+ (Fu et al., 2021) | Adv | ✓ | | | | | | 2.3 | **97.7** $\star$ |
| PERL-AE (Kataria et al., 2021) | Reg | ✓ | | | | | | 4.5 | **95.5** $\star$ |
| Speech Reg. (Polyak et al., 2021) | Gen | ✓ | ✓ | | | | | 4.5 | **95.5** $\star$ |
| SPEC-GAN (Su et al., 2019) | Adv | ✓ | ✓ | | | | | 6.8 | **93.2** $\star$ |
| HiFi-GAN-2 (Su et al., 2021) | Adv | ✓ | ✓ | | | | ✓ | 22.7 | **77.3** $\star$ |
| WSRGlow (Zhang et al., 2021b) | Gen | | | ✓ | | | | 6.8 | **93.2** $\star$ |
| SEANet (Li et al., 2021b) | Adv | | | ✓ | | | | 27.3 | **72.7** $\star$ |
| GSEGAN (Pascual et al., 2019) | Gen | | | ✓ | ✓ | | ✓ | 0.0 | **100.0** $\star$ |
| DNN-S (Mack & Habets, 2019) | Reg | | | | ✓ | | | 2.3 | **92.7** $\star$ |
| CT+TF-UNet (Nair & Koishida, 2021) | Reg | | | | ✓ | ✓ | ✓ | 4.5 | **95.5** $\star$ |
| VoiceFixer (Liu et al., 2021) | Gen | ✓ | ✓ | ✓ | ✓ | | | 29.5 | **70.5** $\star$ |
| UNIVERSE | Gen | ✓ | ✓ | ✓ | ✓ | ✓ | ✓ | n/a | n/a |
| UNIVERSE-Regress | Reg | ✓ | ✓ | ✓ | ✓ | ✓ | ✓ | 11.4 | **88.6** $\star$ |
| UNIVERSE-Denoise | Gen | ✓ | | | | | | 43.2 | **56.8** |
| Ground truth oracle | n/a | ✓ | ✓ | ✓ | ✓ | ✓ | ✓ | **86.4** $\star$ | 13.6 |

significant margin (Table 2, top). In all considered distortions, UNIVERSE is preferred by expert listeners when compared to the corresponding competitor. The closest competitors are HiFi-GAN-2, which considers denoising, dereverb, and equalization, SEANet, which only considers bandwidth extension, and VoiceFixer, which considers denoising, dereverb, bandwidth extension, and declipping. We encourage the reader to listen to the UNIVERSE-enhanced examples in the companion website[1].

## 4.2 VARIATIONS AND INSIGHTS

Another interesting set of results stems from comparing against ablations or ground truth data (Table 2, bottom). In particular, we try to answer the following questions:

1. *Do we find a clear gain from using a generative, diffusion-based approach compared to using classical regression losses?* To answer this question, we performed several preliminary experiments with different regression-based alternatives, and concluded that the best candidate for the subjective test was a version of UNIVERSE with exactly the same configuration and capacity which, instead of a score matching loss for the generator network and diffusion-based sampling, uses MSE and STFT losses for direct waveform prediction (the conditioner network was still found to be superior with the MDN losses). The result of this approach was not at the level of the generative version (Table 2, UNIVERSE-Regress), especially with regard to speech distortion and artifacts. Listeners preferred the generative version 88 % of the time.

2. *Can it be a problem for the model to consider more distortions beyond additive noise?* To assess this, we trained exactly the same version of UNIVERSE with a 500 h train set that consisted of only additive noise mixtures, and evaluated only in the additive noise case. The reason for using 1/3 of the hours used for the universal enhancement case is that we estimate that this number is not far from the amount of real-world additive noise in the full multi-distortion train set. In this case, the results indicate that adding extra data and extra distortions does not affect performance (Table 2, UNIVERSE-Denoise). In fact, there seems to be a slight advantage in doing so (43 vs. 57 % preference), albeit not a significant one.

3. *How far are we from the ideal targets?* To gain intuition about this question, we included recordings from the target speech to the subjective test (with input references featuring all considered distortions). In this case, listeners preferred the ideal targets 86 % of the time (Table 2, Ground truth oracle). This, beyond confirming that listeners were able to spot the clean

---

[1]Link removed to preserve anonymity. Examples are attached to the current submission in a ZIP file. To foster future comparison, we will also provide the first (random) 100 pairs of our validation set on the website.

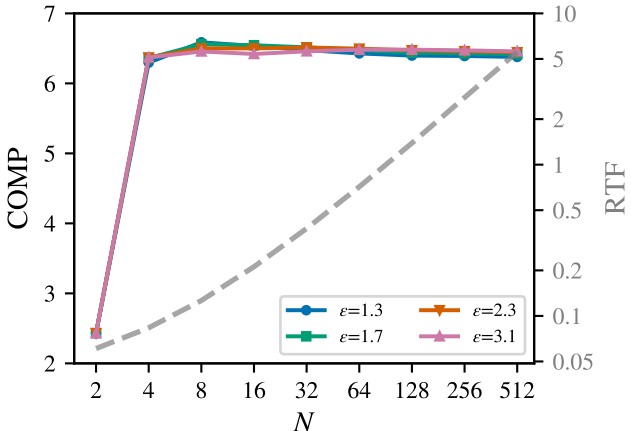

Figure 3: Speed-quality trade-off when varying sampling parameters. Speed is measured by the real-time factor on a Tesla V100 GPU (RTF; gray dashed line) and quality is measured by COMP (colored solid lines). Sampling parameters are the number of iterations $N$ and the constant $\epsilon$ (Sec. 3.2). Results for the other objective metrics are in Appendix E.

references, indicates that there is still some room for improvement for UNIVERSE. Informal listening by the authors indicates that two of the most problematic cases are with very loud noises and strong (and usually long) reverbs. The former tends to yield some babbling while the latter yields noticeable speech distortions and, in some cases, also babbling.

### 4.3 SPEED-QUALITY TRADEOFF

Score-based diffusion models require performing multiple denoising steps for high-quality sampling, and efficient strategies to tackle this issue have been the subject of recent research. In the literature, we find that high-quality sampling can be obtained with relatively few steps for tasks that have a rich conditioning signal such as speech vocoding (for example, less than 10 steps as in Chen et al. (2021); Kong et al. (2021)). Speech enhancement, as formulated here, should also be considered a task with a rich conditioning signal (essentially, low-level speech details are contained at the input, except for the distorted parts). Therefore, we hypothesize that high-quality synthesis can also be achieved with relatively few steps. Our results confirm this hypothesis and show that we can obtain good quality synthesis with as few as 4–8 denoising steps, with a speed above 10 times real-time on GPU (Fig. 3 and Appendix E). Importantly, this holds for a variety of values of $\epsilon$, and without the use of any specific strategy nor any search for an appropriate schedule of $\sigma_t$ (we use plain geometric scheduling).

## 5 CONCLUSION

In this work, we consider the task of speech enhancement as a holistic endeavor, and propose a universal speech enhancer that makes use of score-based diffusion for generation and MDN auxiliary heads for conditioning. We show that this approach, UNIVERSE, outperforms 12 state-of-the-art approaches as evaluated by expert listeners in a subjective test, and that it can achieve high-quality enhancement with only 4–8 diffusion steps. Regarding the potential societal impact, we do not foresee any serious implications at this initial research stage. Despite being a generative model, the nature of the task enforces to enhance what is being input to the system without changing major characteristics, keeping content and intent absolutely unaltered. Thus, for example, the system should not change the speaker's identity or words unless attacked by a third party (we explicitly ask expert listeners to consider identity/word changes in their judgment). Finally, it is also worth noting that data-driven models highly depend on the training data characteristics. Accordingly, before deploying such models, one should ensure that target languages and use cases are sufficiently represented in the train set (we explicitly include multiple languages and recording conditions in our training set).

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

APPENDIX

# A  SCORE-BASED DIFFUSION MODELS

## A.1  THEORY

Diffusion-based models are defined through a forward process where we progressively add noise to samples from the data distribution, $\mathbf{x}_0 \sim p_{\text{data}}$, until we obtain a result that is indistinguishable from a prior tractable distribution, $\mathbf{x}_1 \sim p_{\text{known}}$. In the case of Gaussian noise, $p_{\text{known}}$ also becomes approximately Gaussian, and this process can be modeled (Song et al., 2021) through a stochastic differential equation (SDE):

$$d\mathbf{x} = f(\mathbf{x}, t)dt + g(t)d\mathbf{w}, \tag{2}$$

where $f, g : \mathbb{R} \to \mathbb{R}$ and $\mathbf{w}$ is the standard Wiener process (Brownian motion) indexed by a continuous time variable $t \in [0, 1]$. Different definitions of $f$ and $g$ yield to different but equivalent processes (Song et al., 2021). To model the backward process where we go from $p_{\text{known}}$ to $p_{\text{data}}$, we can then employ the reverse-time SDE (Anderson, 1982)

$$d\mathbf{x} = \left[ f(\mathbf{x}, t) - g(t)^2 \nabla_{\mathbf{x}} \log p_t(\mathbf{x}) \right] dt + g(t)d\bar{\mathbf{w}}, \tag{3}$$

where $\bar{\mathbf{w}}$ is the standard Wiener process in which time flows backward, and $\nabla_{\mathbf{x}} \log p_t(\mathbf{x})$ corresponds to the score of $p_t$, the marginal distribution at time $t$ (Hyvärinen, 2005). Thus, once $f$ and $g$ are defined, in order to obtain $\mathbf{x}_0 \sim p_{\text{data}}$, we only need to know the score function $\nabla_{\mathbf{x}} \log p_t(\mathbf{x})$ to sample $\mathbf{x}_1 \sim p_{\text{known}}$ and simulate the process of Eq. 3.

Since one does not typically have direct access to neither $p_t$ nor $\nabla_{\mathbf{x}} \log p_t(\mathbf{x})$, the solution is to approximate the latter with a neural network $S(\mathbf{x}, t)$. In order to train $S$, Vincent (2011) showed that, for a given $t$, minimizing the score matching objective

$$\mathcal{L} = \mathbb{E}_{\mathbf{x}_t} \left[ \frac{1}{2} \left\| S(\mathbf{x}_t, t) - \nabla_{\mathbf{x}} \log p_t(\mathbf{x}) \right\|_2^2 \right]$$

is equivalent to minimizing the denoising objective

$$\mathcal{L} = \mathbb{E}_{\mathbf{x}_t | \mathbf{x}_0} \mathbb{E}_{\mathbf{x}_0} \left[ \frac{1}{2} \left\| S(\mathbf{x}_t, t) - \nabla_{\mathbf{x}} \log p_t(\mathbf{x}_t | \mathbf{x}_0) \right\|_2^2 \right],$$

where $p_t(\mathbf{x}_t | \mathbf{x}_0)$ corresponds to a Gaussian kernel (Song et al., 2021; Särkkä & Solin, 2019), the transition kernel for the forward SDE (Eq. 2). For all $t$, one can use the continuous generalization (Song & Ermon, 2019)

$$\mathcal{L} = \mathbb{E}_t \mathbb{E}_{\mathbf{x}_t | \mathbf{x}_0} \mathbb{E}_{\mathbf{x}_0} \left[ \frac{\lambda_t}{2} \left\| S(\mathbf{x}_t, t) - \nabla_{\mathbf{x}} \log p_t(\mathbf{x}_t | \mathbf{x}_0) \right\|_2^2 \right] \tag{4}$$

with $t \sim \mathcal{U}(0, 1)$, where $\lambda_t$ is an appropriately chosen weight that depends on $t$ (Ho et al., 2020; Song & Ermon, 2020).

Sampling with score-based diffusion models is done by simulating or solving the reverse-time SDE (Eq. 3) with a finite (discrete) time schedule. This can be done in many ways, for instance by using numerical SDE and ODE solvers, as introduced by Song et al. (2021). Other schemes that have shown competitive performance include predictor-corrector schemes (Song et al., 2021), ancestral sampling (Ho et al., 2020), and variations of Langevin dynamics (Song & Ermon, 2020; Jolicoeur-Martineau et al., 2021).

## A.2  IN PRACTICE

In our work, we employ the so-called variance exploding schedule (VE; Song et al., 2021), which corresponds to choosing

$$f(\mathbf{x}, t) = 0 \qquad \text{and} \qquad g(t) = \sqrt{\frac{d\sigma^2(t)}{dt}} \tag{5}$$

in Eq. 2. Then, the associated transition kernel for the forward process (Särkkä & Solin, 2019) is $p_t(\mathbf{x}_t | \mathbf{x}_0) = \mathcal{N} \left( \mathbf{x}_t; \mathbf{x}_0, [\sigma_t^2 - \sigma_0^2]\mathbf{I} \right)$, which in practice is approximated by

$$p_t(\mathbf{x}_t | \mathbf{x}_0) \approx \mathcal{N} \left( \mathbf{x}_t; \mathbf{x}_0, \sigma_t^2 \mathbf{I} \right) \tag{6}$$

since $\sigma_0 \to 0$ (see also Song et al., 2021). The intuition is that $p_{t=0}$ becomes indistinguishable from $p_{\text{data}}$, and that $p_{t=1}$ becomes indistinguishable from $p_{\text{known}}$ (a Gaussian distribution). In other words, perturbation and signal should become imperceptible at $t = 0$ and $t = 1$, respectively. In the VE schedule, the scale of the signal $\mathbf{x}_0$ is kept intact, and then it corresponds to $g(t)$ to fulfill that notion through a variance schedule (see Eq. 5). Therefore, $\sigma_0$ should be negligible while $\sigma_1$ should be large, compared to the variability of $\mathbf{x}_0$. That is, $\sigma_0^2 \ll \mathbb{E}[(\mathbf{x}_0 - \mathbb{E}[\mathbf{x}_0])^2] \ll \sigma_1^2$.

The use of the approximated transition kernel (Eq. 6) implies that $\mathbf{x}_t = \mathbf{x}_0 + \sigma_t \mathbf{z}_t$, $\mathbf{z}_t \sim \mathcal{N}(\mathbf{0}, \mathbf{I})$, and that

$$-\nabla_{\mathbf{x}_0} \log p_t(\mathbf{x}_t | \mathbf{x}_0) \approx -\nabla_{\mathbf{x}_0} \left[ C - \frac{(\mathbf{x}_t - \mathbf{x}_0)^2}{2\sigma_t^2} \right] = \frac{\mathbf{x}_t - \mathbf{x}_0}{\sigma_t^2},$$

where $C$ is a constant (see also Vincent, 2011)). Substituting these into Eq. 4 and operating yields

$$\mathcal{L} = \mathbb{E}_t \mathbb{E}_{\mathbf{z}_t} \mathbb{E}_{\mathbf{x}_0} \left[ \frac{\lambda_t}{2} \left\| S(\mathbf{x}_0 + \sigma_t \mathbf{z}_t, t) + \frac{\mathbf{z}_t}{\sigma_t} \right\|_2^2 \right].$$

From Eq. 5, it now remains to set how $\sigma$ evolves. Song & Ermon (2019; 2020) choose a geometric progression for $\sigma_t$,

$$\sigma_t = \sigma_{\min} \left( \frac{\sigma_{\max}}{\sigma_{\min}} \right)^t,$$

and provide some generic guidance on how to select $\sigma_{\min}$ and $\sigma_{\max}$. They also justify setting $\lambda$ proportional to the noise variance at time $t$: $\lambda_t = \sigma_t^2$. Using this weighting yields

$$\mathcal{L} = \mathbb{E}_t \mathbb{E}_{\mathbf{z}_t} \mathbb{E}_{\mathbf{x}_0} \left[ \frac{1}{2} \| \sigma_t S(\mathbf{x}_0 + \sigma_t \mathbf{z}_t, t) + \mathbf{z}_t \|_2^2 \right].$$

Finally, instead of using $t$ directly as input for $S$, we use $\sigma_t$ and train $S$ with a continuum of noise scales (Chen et al., 2021). In addition, we need to use some conditioning information $\mathbf{c}$ as an indication of what to generate (in our case, a signal derived from the distorted speech $\tilde{\mathbf{x}}$). With that, our training loss becomes

$$\mathcal{L} = \mathbb{E}_t \mathbb{E}_{\mathbf{z}_t} \mathbb{E}_{\mathbf{x}_0} \left[ \frac{1}{2} \| \sigma_t S(\mathbf{x}_0 + \sigma_t \mathbf{z}_t, \mathbf{c}, \sigma_t) + \mathbf{z}_t \|_2^2 \right].$$

This is the loss denoted as $\mathcal{L}_{\text{SCORE}}$ in the main paper.

For sampling with the VE schedule, we resort to consistent annealed sampling (Jolicoeur-Martineau et al., 2021). We use the discretization of $t \in [0, 1]$ into $N$ uniform steps $t_n = (n - 1)/(N - 1)$, $n = \{1, \ldots N\}$, which implies discretized progressions for $\mathbf{x}_t$ and $\sigma_t$. Starting at $n = N$, we initialize $\mathbf{x}_{t_N} = \sigma_{t_N} \mathbf{z}_{t_N}$, and then recursively compute

$$\mathbf{x}_{t_{n-1}} = \mathbf{x}_{t_n} + \eta \sigma_{t_n}^2 S(\mathbf{x}_{t_n}, \mathbf{c}, \sigma_{t_n}) + \beta \sigma_{t_{n-1}} \mathbf{z}_{t_{n-1}},$$

for $n = N, N - 1, \ldots 2$. Finally, at $n = 1$ ($t_1 = 0$), we take

$$\bar{\mathbf{x}}_0 = \mathbf{x}_0 + \sigma_0^2 S(\mathbf{x}_0, \mathbf{c}, \sigma_0),$$

which corresponds to the empirically denoised sample (Jolicoeur-Martineau et al., 2021). The values of $\eta$ and $\beta$ are determined through the parameterization (Serrà et al., 2021a)

$$\eta = 1 - \gamma^\epsilon, \qquad \beta = \sqrt{1 - \left( \frac{1 - \eta}{\gamma} \right)^2},$$

where $\gamma \in (0, 1)$ is the ratio of the geometric progression of the noise, $\gamma = \sigma_{t_n}/\sigma_{t_{n+1}}$, and the constant $\epsilon \in [1, \infty)$ is left as a hyper-parameter. Note that the value of $\gamma$ changes with the chosen number of discretization steps $N$, thus facilitating hyper-parameter search for continuous noise scales at different $N$ (Serrà et al., 2021a).

## B  IMPLEMENTATION DETAILS

**Architecture overview** — The architecture contains two main parts that are jointly trained: the generator network, which is tasked with estimating the score of the perturbed speech distributions, and the conditioner network, which creates the conditioning signal required for synthesizing speech with preserved speaker characteristics. Both the generator and the conditioner networks are essentially encoder-decoder architectures, enhanced with various conditioning signals and skip connections (see figure in main paper). The encoding and decoding is done by down- and up-sampling, respectively, using convolutional blocks, which are the main building blocks of UNIVERSE. Throughout the architecture, the multi-parametric rectified linear unit (PReLU) activation function is used prior to all the layers (except where we specify otherwise), and all convolutional layers are one dimensional. Unless stated otherwise, we use PyTorch's defaults (Paszke et al., 2019, version 1.10.1).

**Convolutional block** — The convolutional block consists of a core part which is common between all blocks, and an optional prior or posterior part that can exist depending on the block's functionality (Fig. 4-B). The core part contains a convolutional layer of kernel width 5, followed by two convolutional layers of kernel width 3, and we use a residual sum between its input and output. All residual sums and skip connections are weighted by $1/\sqrt{r}$, where $r$ is the number of elements in the sum.

If the convolutional block will be used for up-sampling, the core's input is processed initially by a transposed convolutional layer of kernel width and stride equal to the up-sampling ratio, without padding (Pons et al., 2021). On the contrary, if the block will be used for down-sampling, the output of the core is processed by a strided convolutional layer of kernel width and stride equal to the down-sampling ratio, also without padding. Every down-sampling operation is accompanied by a channel expansion of factor 2, and up-sampling by a reduction of 2. We start with 32 channels in the encoder and have 512 in the latent. Depending on a block's position in the architecture we use skip connections and a maximum of two types of conditioning signals, all injected to a dedicated part in the convolutional block (Fig. 4-B). If a signal will be extracted with a skip connection, we take it from the output of the residual sum, and insert it to the target block before the residual signal.

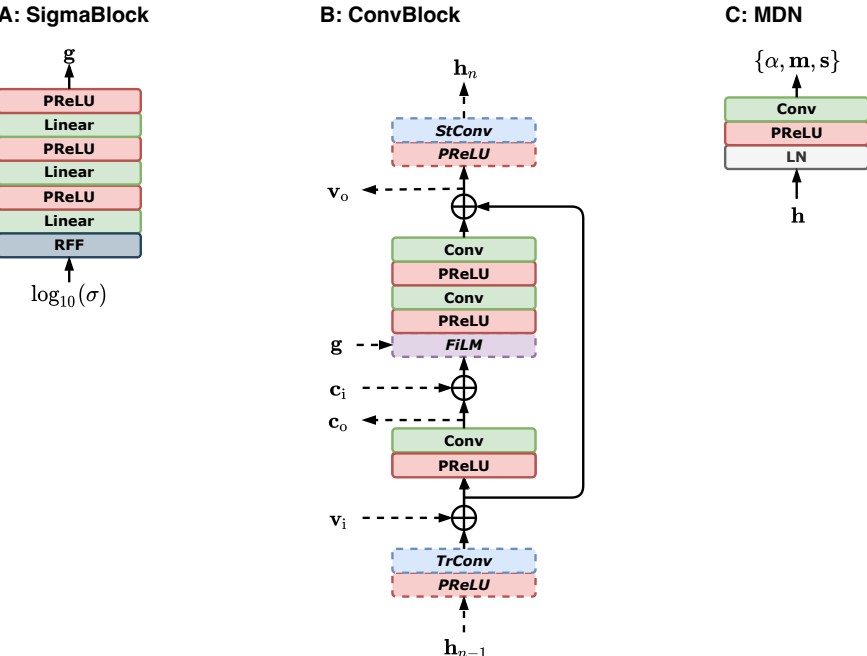

Figure 4: Diagram of the individual blocks not depicted in the main paper. Dashed connections and blocks are optional, depending on the functionality of the block.

In order to preserve speaker characteristics that are consistent throughout a frame, we use conditioning signals between the generator and the conditioner. These conditioning signals are taken from the output of the first convolutional layer of the core, and inserted to the same position in the target block's core. Furthermore, a secondary conditioning signal containing information about the noise level is injected to all the convolutional blocks of the generator. This conditioning is done with FiLM (Perez et al., 2018).

**Conditioner network —** The conditioner network contains a stem cell, a down-sampling encoder, a recursive middle part, and an up-sampling decoder. Both the encoder and decoder are made up of convolutional blocks that perform only down-sampling and up-sampling, respectively. The distorted signal is down-sampled to a 100 Hz sampling rate, with the encoder following a {2,4,4,5} factor progression, recursively processed in the middle part, and up-sampled to the original sampling rate with the decoder, mirroring the encoder's progression. At each encoder block, we use skip connections, process them with adaptor networks, and sum the adapted signals with the output of the last encoder block. The adaptor networks consist of strided convolutions in order to match the sampling rate of the summation inputs.

Alongside this main path, we extract and normalize 80 mel band features from the distorted signal, process them with a convolutional layer followed by a convolutional block, and add the result to the summation of the down-sampled signals. The hop size is equal to the total down-sampling rate, preserving the sampling rate of the features during processing. We process the feature-enriched signal with the conditioner's middle part, consisting of a convolutional block, a two-layer bi-directional GRU with a residual sum, and a final convolutional block, all of them preserving the sampling rate. We finally up-sample the latent representation to the original sampling rate with the decoder, using an initial convolutional block preserving the sampling rate and numerous blocks performing up-sampling. We obtain a hierarchical conditioning for the generator network by taking conditioning signals from each block of the decoder, corresponding to all the sampling rates. In order to adapt these representations to the ones of the generator network, we process each conditioning signal with linear layers.

**Generator network —** The generator network is also an encoder-decoder architecture with UNet-like skip connections, where there is a direct connection between each encoder and decoder block pair sharing the same sampling rate. We use the same down- and up-sampling progression as in the conditioner network, hence achieving parallel up-sampling blocks between the generator and conditioner decoders. However, we limit the middle part to only a single bi-directional GRU layer without residual sums. The output of the final block is processed by a convolutional layer to reduce its dimension back to the score's dimension.

The generator is conditioned on two signals: Fourier features depending on the current noise level and the hierarchical conditioning extracted from the distorted speech. For the former, we embed the noise level with logarithmic compression and use it to modulate frequencies of random Fourier features (RFFs) sampled from a standard normal distribution. Following the extraction of RFFs, we process them with a fully connected network of 3 times repeated linear layers followed by PReLUs to obtain noise level embeddings. This block, which we call SigmaBlock, expands 32 pairs of Fourier coefficients into 256 channels. (Fig. 4-A). After the embeddings are extracted, they are adapted to each block's frame rate and dimension with a linear projection layer. The conditioning signals taken from the conditioner network's decoder are injected to the generator network's decoder at each block with the same sampling rate, using summation. This way, the estimated score is conditioned with multiple sampling rates starting from the lowest (100 Hz), until the original (16 kHz).

**Training objective —** The main loss is the denoising score matching loss outlined in the main paper and derived in Appendix A.2. Additionally, we introduce auxiliary losses in order to structure the conditioner network's latent representation and prime its final block's output to perform speech enhancement. We use separate mixture density networks (MDN; Bishop, 1994) to model the distributions of the clean signal waveform and features. Each feature and its delta distribution is jointly estimated with an MDN that models the distribution using 3 multivariate Gaussian components with diagonal covariance. The parameters of the mixture components are learned with convolutional layers of kernel width 3 and layer normalization (Fig. 4-C). We calculate the negative log-likelihood (NLL) of the feature MDNs and average them. Then, the total objective is to minimize the sum of the denoising score matching loss, waveform NLL, and feature NLLs. Here, we would like to underline

that auxiliary losses are generative, as they model continuous data distributions, making UNIVERSE a fully-generative model (that is, not using any classification or regression losses). Moreover, as the auxiliary losses are taken out of the main signal path, we do not need any sampling procedure neither in training nor in validation stages.

**Optimization —** The resulting model contains about 49 M parameters. We train it with a batch size of 32, where each sample is an approximately two-seconds long frame with 16 kHz sampling rate, and using automatic mixed precision. We apply 1 M parameter updates, which takes approximately four and a half days using 2 Tesla V100 GPUs in parallel. The large/final model has 189 M parameters, is trained for 2.5 M iterations using a batch size of 64, and takes less than 14 days using 8 Tesla V100 GPUs. We use the Adam optimizer with a maximum learning rate of $2 \cdot 10^{-4}$ and schedule the learning rate with a cosine scheduler. The schedule includes a warm-up of 50 k iterations, increasing the learning rate from a starting point $1.6 \cdot 10^{-6}$. We additionally implement a manual weight decay, excluding biases and the PReLU weights, and set its coefficient to 0.01.

## C  DATA AND DISTORTIONS

To create our training data, we sample chunks of 3.5–5.5 seconds from speech recordings of an internal pool of data sets, featuring multiple speakers, languages, prosody/emotions, and diverse recording conditions. Speech is pre-selected to be clean, but nonetheless can contain minimal background noise and reverb. We down-sample to 16 kHz mono and follow two different signal paths to create input and target pairs. With distortions that introduce delays we automatically time-align at the sample level using a custom correlation function.

**Input —** To programmatically generate distorted versions, we randomly choose the number of distortions between $\{1, 2, 3, 4, 5\}$, with probabilities $\{0.35, 0.45, 0.15, 0.04, 0.01\}$, respectively. Next, we cascade distortion types with random parameters (bounds for distortion parameters are selected such that distortions are perceptually noticeable by expert listeners; random selection is typically uniform, except for parameters that have an intuitive logarithmic behavior, such as frequency, in which case we sample uniformly in logarithmic space). Table 3 summarizes the distortion families and types we consider, together with their weights (distortion type weights will define the probability of selecting each distortion type). In total, we consider 55 distortions, which we group into 10 families.

The ranges we consider for the distortion parameters include: SNRs between $-5$ and 25 dB for all added noises, slopes between 0.01 and 0.7 for colored noise, DC components between $10^{-6}$ and $10^{-1}$, non-stationary noise lengths between 20 and 350 ms, frequency ranges between 100 and 7500 Hz for filters and tones (except low and high pass, with a minimum and maximum of 1 kHz, respectively), Qs between 0.1 and 2 for filters, gains between $-12$ and 6 dB for filters and random equalization, between 2 and 20 mel bands for random equalization, gap lengths between 20 and 80 ms, temporal probabilities between 0.5 and 3 times/sec for transmission distortions, percentiles between 0.5 and 50 for clipping, resample frequencies of 4, 6, 8, 11, 12, and 14 kHz, wet proportions between 0.2 and 1 for reverbs, augmentation of early and tail reflections for room impulse responses, room sizes between 3 and 1000 m$^2$ for algorithmic reverbs, bit rates between 2 and 96 kbps for codecs, compression ratios between 2 and 10, window lengths between $2^9$ and $2^{12}$ for spectral manipulations, amounts between 0.5 and 0.8 for spectral manipulation probability, wet proportions between 0.1 and 0.75 for other effects, and linear gains between 0 and 1 for plosive and sibilance addition.

**Target —** Given that the design of UNIVERSE does not rely on assumptions regarding the nature of the distortions (for instance, a good example would be the assumption that noise is additive), nothing prevents us from using a different target than the original signal that was used as input to the distortion chain. In addition, we want to ensure that the generated signal is of the top/highest quality, which is a characteristic that is not shared by some of the original clean speech signals that we sample. Therefore, we decide to apply a small enhancement chain to clean the signals to be used as target. In particular, we apply four consecutive steps[2]:

1. Denoising — We employ a denoiser to remove any (minimal) amount of noise that might be present in the recording. Since the pre-selected speech is already of good quality, we expect that

---

[2]All of these algorithms are available at `https://dolby.io/`

Table 3: Considered distortion types, grouped by family.

| ID | Family | Type | Weight | Randomized parameters |
|---|---|---|---|---|
| 1 | Band limiting | Band pass filter | 5 | Frequencies, filter characteristics. |
| 2 | | High pass filter | 5 | Frequency, filter characteristics. |
| 3 | | Low pass filter | 20 | Frequency, filter characteristics. |
| 4 | | Down-sample | 30 | Frequency, method. |
| 5 | Codec | AC3 codec | 2 | Bit rate, codec configuration. |
| 6 | | EAC3 codec | 3 | Bit rate, codec configuration. |
| 7 | | MDCT codec | 15 | Bit rate, codec configuration. |
| 8 | | MP2 codec | 5 | Bit rate, codec configuration. |
| 9 | | MP3 codec | 20 | Bit rate, codec configuration. |
| 10 | | Mu-law quantization | 3 | Mu. |
| 11 | | OGG/Vorbis codec | 3 | Bit rate, codec configuration. |
| 12 | | OPUS codec 1 | 15 | Bit rate, codec configuration. |
| 13 | | OPUS codec 2 | 2 | Bit rate, codec configuration. |
| 14 | Distortion | More plosiveness | 10 | Gain. |
| 15 | | More sibilance | 10 | Gain. |
| 16 | | Overdrive | 5 | Gain, harmonicity. |
| 17 | | Threshold clipping | 8 | Gain. |
| 18 | Loudness dynamics | Compressor | 10 | Ratio, compressor characteristics. |
| 19 | | Destroy levels | 20 | Gains, durations, temporal probability. |
| 20 | | Noise gating | 10 | Gate characteristics. |
| 21 | | Simple compressor | 3 | Ratio. |
| 22 | | Simple expansor | 2 | Ratio. |
| 23 | | Tremolo | 2 | Tremolo characteristics. |
| 24 | Equalization | Band reject filter | 5 | Frequencies, filter characteristics. |
| 25 | | Random equalizer | 15 | Number of bands, gains. |
| 26 | | Two-pole filter | 10 | Frequency, filter characteristics. |
| 27 | Recorded noise | Additive noise | 150 | SNR, noise type (cafeteria, traffic, nature, classroom, keyboard, plane, music, ...). |
| 28 | | Impulsional additive noise | 30 | SNR, noise type, temporal probability. |
| 29 | Reverb/delay | Algorithmic reverb 1 | 30 | Gain, reverb characteristics. |
| 30 | | Algorithmic reverb 2 | 5 | Gain, reverb characteristics. |
| 31 | | Chorus | 1 | Gain, chorus characteristics. |
| 32 | | Phaser | 1 | Gain, phaser characteristics. |
| 33 | | RIR convolution | 120 | Gain, room impulse response (RIR), augment. |
| 34 | | Very short delay | 3 | Gain, delay time. |
| 35 | Spectral manipulation | Convolved spectrogram | 1 | Window, amount. |
| 36 | | Griffin-Lim | 3 | Window, amount. |
| 37 | | Phase randomization | 1 | Window, amount. |
| 38 | | Phase shuffle | 1 | Window, amount. |
| 39 | | Spectral holes | 1 | Window, amount. |
| 40 | | Spectral noise | 1 | Window, amount. |
| 41 | Synthetic noise | Colored noise | 15 | SNR, slope. |
| 42 | | DC component | 1 | Amplitude. |
| 43 | | Electricity tone | 6 | SNR, frequency, type of waveform. |
| 44 | | Non-stationary colored noise | 5 | SNR, slope, duration, temporal probability. |
| 45 | | Non-stationary DC component | 1 | Amplitude, duration, temporal probability. |
| 46 | | Non-stationary electricity tone | 3 | SNR, frequency, duration, temporal prob. |
| 47 | | Non-stationary random tone | 1 | SNR, frequency, duration, temporal prob. |
| 48 | | Random tone | 2 | SNR, frequency, type of waveform. |
| 49 | Transmission | Frame shuffle | 10 | Length, temporal probability. |
| 50 | | Insert attenuation | 3 | Length, temporal probability, gain. |
| 51 | | Insert noise | 5 | Length, temporal probability, SNR. |
| 52 | | Perturb amplitude | 1 | Length, temporal probability, gain. |
| 53 | | Sample duplicate | 2 | Length, temporal probability. |
| 54 | | Silent gap (packet loss) | 15 | Length, temporal probability. |
| 55 | | Telephonic speech | 10 | Frequencies, compression ratio, filter type. |

any decent denoiser will have no problems and will not introduce any noticeable distortion. We carefully verified that by listening to several examples.

2. Deplosive — We employ a deplosive algorithm, which is composed of detection and processing stages. The processing acts only on the level of the plosive bands.

3. Deesser — We employ a basic deesser algorithm, which is composed of detection and processing stages. The processing acts only on the level of the sibilant bands.

4. Dynamic EQ — We also employ a signal processing tool that aims at bringing the speech to a predefined equalization target. The target is level-dependent, so that the algorithm can act as a compressor and/or expander depending on the characteristics of the input speech.

Interestingly, passing the original clean recordings through this chain forces the model to not rely on bypassing input characteristics (especially low-level characteristics), which we find particularly relevant to remove other non-desirable input characteristics. In addition, the chain provides a more homogeneous character to the target speech, which should translate into some characteristic 'imprint' or 'signature sound' of UNIVERSE.

## D    EVALUATION

### D.1    OBJECTIVE MEASURES

We report results with COVL (Loizou, 2013) and STOI (Taal et al., 2010) as computed by the pysepm[3] package using default parameters (version June 30, 2021). WARP-Q (Jassim et al., 2021) is used also with default parameters[4] (version March 14, 2021) and SESQA is a reference-based version of the reference-free speech quality measure presented in Serrà et al. (2021b), trained on comparable data and losses as explained by the authors. These four objective measures are only used to aid in the assessment of the development steps presented in the main paper. PESQ (Rix et al., 2001), COVL, and STOI are also used to compare with the state of the art in the task of speech denoising in Table 4 below. Calculation of PESQ is also done using pysepm[5].

### D.2    SUBJECTIVE TEST

We perform a subjective test considering 15 competitor approaches (12 existing ones plus 3 ablations of the proposed approach, as reported in the main paper). For each existing approach, we download distorted and enhanced pairs of speech recordings from the corresponding demo websites, and use those as the main materials for the test. We randomly select at least 25 pairs per system, and remove a few ones for which we consider the distorted signal is too simple/easy to enhance (for instance, signals that have a very high SNR when evaluating a denoising system, almost no clipping when evaluating a de-cplipping system, and so forth). This is done to reduce potentially ambivalent results in the test, as the two systems would presumably perform very well and the listener would be confused on which to choose (removed pairs were never more than 5 per approach). For a few systems that do not have enough material on their demo page, we download the code and enhance a random selection of distorted input signals from other systems. That is the case of Demucs, MetricGAN+, and VoiceFixer. For the ablations of the proposed approach, we take the first (random) 25 pairs of our validation set. In total, we have over $(12+3) \times 25$ enhanced signals from which to judge preference. All distorted signals are enhanced by UNIVERSE using $N = 64$ diffusion steps and hyper-parameter $\epsilon = 2.3$.

A total of 22 expert listeners voluntarily participated in the test. The test features a set of instructions and then shows a list of triplets from which to perform judgment (Fig. 5). Every time a listener lands on the test page, triplets of signals are uniformly randomly selected for each approach: input-distorted signal (Input), competitor-enhanced signal (A or B), and UNIVERSE-enhanced signal (A or B). The order of A or B, as well as the order of the triplets is also uniformly randomly selected when landing to the web page. Preference for an enhanced signal can only be A or B (forced-choice test). Every subject listens to two triplets per competitor system, performing a total of 30 preference choices for 15 approaches (two per approach, yielding a total of $22 \times 2$ preference judgments per system).

The results of the test are based on counting preference choices per competitor approach (% of preference). Statistical significance is determined with a binomial test using $p < 0.05$ and the Holm-Bonferroni adjustment to compensate for multiple comparisons. In the results table, we also

---

[3]https://github.com/schmiph2/pysepm
[4]https://github.com/wjassim/WARP-Q
[5]We always use wide-band PESQ (also for computing COVL). The only exception are the results of Table 4, where we report both narrow- and wide-band PESQ (COVL is still computed with wide-band PESQ).

Figure 5: Screenshot of the subjective test interface. On the top, instructions are given to expert listeners. On the bottom, random test triplets are provided to the listeners. For every test, subjects listen to two triplets comparing UNIVERSE with an existing approach, and enter preference for 15 of such approaches.

display which distortions were tackled by the competitor approach. Note that, since testing materials have different distortions and contents per approach, one cannot infer a ranking of systems based on preference % (that is, one cannot compare preferences between rows of the table; actually they even correspond to different tasks: denoising, de-clipping, restoring codec artifacts, etc.). Preference % only represents a pairwise comparison between UNIVERSE and the corresponding competitor approach listed on the left of the table.

# E  ADDITIONAL RESULTS

## E.1  DENOISING TASK WITH OBJECTIVE METRICS

To have a comparison on a well-established benchmark, we can consider the task of speech denoising, which has a long tradition in the speech community and, in the last years, has seen a consolidation of test data sets and objective metrics (this is something that, to the best of our knowledge, has not yet happened with other tasks like dereverberation, declipping, bandwidth extension, and so on). Two widely-used data sets with an established test partition are Voicebank-DEMAND (Valentini-Botinhao, 2017) and IS20 DNS-Challenge (Reddy et al., 2020). Since the standard objective metrics are PESQ, COVL, and STOI, and given that UNIVERSE produces audio with further enhancements like leveling/dynamics or a specific equalization, we need to train a new version of UNIVERSE specifically for the denoising task. This is important because, otherwise, UNIVERSE would be performing additional enhancements besides denoising and, more importantly, the standard objective metrics would not find an agreement between the ground truth and the estimated clean speech, what would result in the whole evaluation being unfair to UNIVERSE.

Table 4: Comparison with the state of the art on the speech denoising task using objective metrics (for all metrics, the higher the better). The first block of the table contains results for existing generative approaches (upper part), while the second block of the table contains results for regression/adversarial approaches (middle part). Bottom rows correspond to the proposed approach UNIVERSE.

| Approach | VoiceBank-DEMAND | | | | IS20 DNS Challenge (no rev) | | | |
|---|---|---|---|---|---|---|---|---|
| | $\text{PESQ}_{\text{NB}}$ | PESQ | COVL | STOI | $\text{PESQ}_{\text{NB}}$ | PESQ | COVL | STOI |
| SEGAN (Pascual et al., 2017) | | 2.16 | 2.80 | | | | | |
| DSEGAN (Phan et al., 2020) | | 2.39 | 2.90 | 0.93 | | | | |
| SE-Flow (Strauss & Edler, 2021) | | 2.43 | 3.09 | | | | | |
| DiffuSE (Lu et al., 2021) | | 2.44 | 3.03 | | | | | |
| CDiffuSE (Lu et al., 2022) | | 2.52 | 3.10 | | | | | |
| PR-WaveGlow (Maiti & Mandel, 2020) | | | 3.10 | 0.91 | | | | |
| CycleGAN-DCD (Yu et al., 2021) | | 2.90 | 3.49 | 0.94 | | | | |
| PSMGAN (Routray & Mao, 2022) | | 2.92 | 3.52 | | | | | |
| PoCoNet (Isik et al., 2020) | | | | | | 2.75 | 3.42 | |
| FullSubNet (Hao et al., 2021) | | | | | 3.31 | 2.78 | | 0.96 |
| DCCRN+ (Lv et al., 2021) | | 2.84 | | | 3.33 | | | |
| CTS-Net (Li et al., 2021a) | | 2.92 | 3.59 | | 3.42 | 2.94 | | 0.97 |
| Demucs (Défossez et al., 2020) | | 3.07 | 3.63 | 0.95 | | | | |
| SN-Net (Zheng et al., 2021) | | 3.12 | 3.60 | | 3.39 | | | |
| SE-Conformer (Kim & Seo, 2021) | | 3.12 | 3.82 | 0.95 | | | | |
| MetricGAN+ (Fu et al., 2021) | | 3.15 | 3.64 | | | | | |
| PERL-AE (Kataria et al., 2021) | | 3.17 | 3.83 | 0.95 | | | | |
| HiFi-GAN-2 (Su et al., 2021) | | 3.18 | 3.84 | | | | | |
| Loss Mixup (Chang et al., 2021) | | 3.26 | 3.91 | | | | | |
| DPT-FSNet (Dang et al., 2022) | | **3.33** | **4.00** | **0.96** | | 3.26 | | **0.98** |
| UNIVERSE-Denoise | 3.85 | 3.21 | 3.68 | 0.95 | 3.58 | 3.01 | 3.60 | 0.97 |
| UNIVERSE-Denoise-$\mathbb{E}$ | **3.94** | **3.33** | 3.82 | **0.96** | **3.73** | 3.17 | **3.75** | **0.98** |

In addition, because these standard metrics are known to disfavor generative approaches due to lack of waveform alignment or minor spectral nuances (see Jassim et al., 2021, for a discussion and further pointers), we need to devise a method to produce outputs that are 'less generative' and can therefore better coincide with what standard metrics measure. Inspired by Fejgin et al. (2020), we decide to use an expectation of the enhanced waveform. Intuitively, that expectation should minimize to a certain extent different nuances introduced by the generative model, especially regarding small misalignments and minor spectral nuances. To compute such expectation, we sample 10 times using the same conditioning (distorted) signal, and take the sample average of the resulting waveforms. While this has some audible effect such as lowering the presence of high frequencies, it clearly shows a boost in the standard objective metrics, probably because they focus more on small misalignments and low frequencies. We use $\mathbb{E}$ to denote this version of the approach. We also want to stress that we solely use this expectation version for the result in the corresponding row of Table 4.

Table 4 shows the results for the denoising task. In it, we can observe the difference in standard metrics between generative and regression/adversarial approaches (first two blocks of the table). The best-performing approaches in the generative block struggle to get the numbers of the worse-performing approaches in the regression/adversarial block (by listening to some examples from both blocks, we believe this is a metrics issue since we could not find a clear perceptual difference). Interestingly, we observe that even the generative version of the proposed approach (UNIVERSE-Denoise) clearly surpasses all existing generative approaches in terms of standard metrics. In addition, the 'less generative' variant using an expectation over multiple realizations (UNIVERSE-Denoise-$\mathbb{E}$) shows a clear improvement for those metrics, with values that become competitive with the non-generative state-of-the-art. The results of our subjective test, which include some of the competing approaches in Table 4, suggest that UNIVERSE outperforms those in such more realistic evaluation.

## E.2 SPEED-QUALITY TRADE-OFF

For completeness, we provide the speed-quality plots for every considered objective metric in Fig. 6. As in the main paper, synthesis parameters are the number of denoising iterations $N$ and the hyper-parameter $\epsilon$ (Sec. A.2). The real-time factor (RTF) is defined as the time to process a recording using

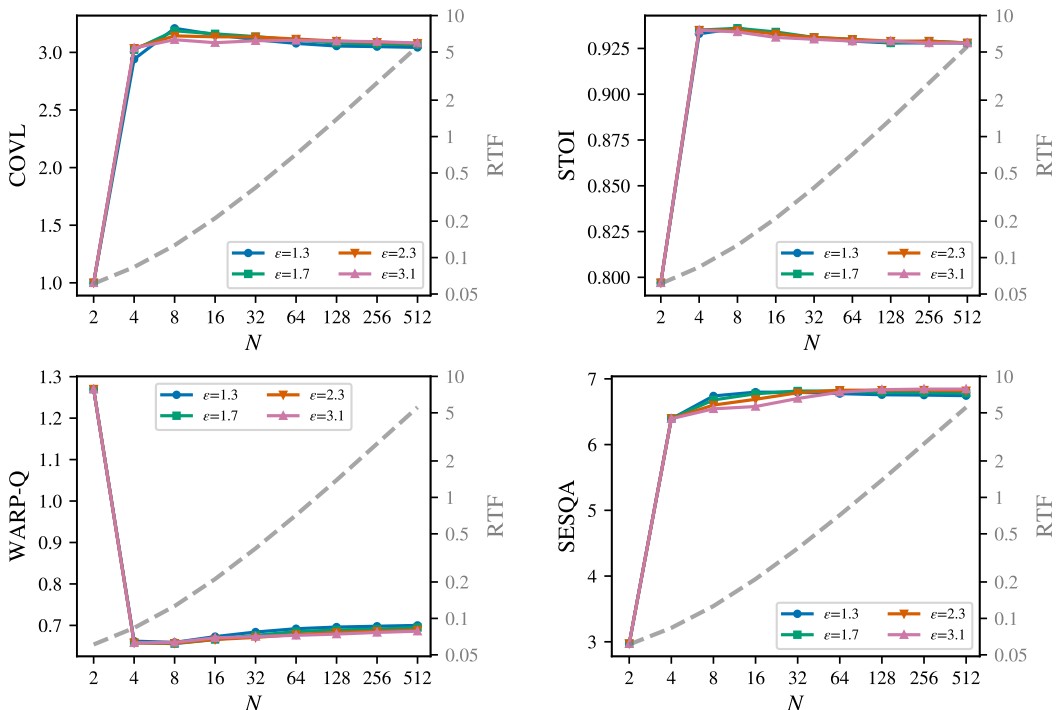

Figure 6: Speed-quality trade-off when varying synthesis parameters. Speed is measured by the real-time factor on a Tesla V100 GPU (RTF; gray dashed line) and quality is measured with the considered objective metrics on the validation set (colored solid lines).

a single Tesla V100 GPU divided by the duration of that recording (for instance, if UNIVERSE takes 2 seconds for enhancing a 20-second recording, RTF=0.1 and we say it is 10 times faster than real time).

