# OpenReview forum: "Universal Speech Enhancement with Score-based Diffusion"
_ICLR.cc/2023/Conference — Submitted to ICLR 2023_

### Official Review · Reviewer_XfqC · 2022-10-15

**Confidence:** 4
**Correctness:** 3
**Technical Novelty And Significance:** 1
**Empirical Novelty And Significance:** 2
**Recommendation:** 5

**Clarity, Quality, Novelty And Reproducibility:**

The paper is written clearly. While an impressive feat of engineering, the paper's contributions are neither novel nor reproducible.

**Strength And Weaknesses:**

Strengths:
- To the best of my knowledge, this paper is the first in the speech enhancement literature to handle this many speech distortions. 55 distortions from 10 different families are considered: bandwidth reduction, clipping, codec artifacts, silent gaps, excessive dynamics
compression/expansion, sub-optimal equalization, noise gating, and others.
- While it is not the first paper that proposes to use diffusion for speech enhancement, the authors show that it performs better than existing approaches on subjective side-by-side evaluations, as well as automated metrics on the VoiceBank-Demand test set.

Weaknesses:
- While an impressive engineering feat, the paper contributes little to the community's theoretical or empirical understanding of speech enhancement or diffusion.
- The findings in the paper are also generally not reproducible. The authors use internal training data and programmatically generate distorted speech, but neither the training data nor code for the programmatic distortions are made available. The paper proposes a very "hacky" U-Net architecture with numerous non-trivial and non-standard modifications, but while the ablation study is useful, it is hard to see how others can easily replicate this without the authors releasing the code for it. Finally, the side-by-side evaluations are also subjective and not generally reproducible. I think it is a huge missed opportunity, because the authors could have taken the chance to establish a new benchmark task and associated metrics for "universal" speech enhancement.



**Summary Of The Paper:**

This paper proposes a "universal" speech enhancement model that handles 55 different distortions including background noise, reverberation, codec artifacts, etc. The model uses a UNet-like generator that has been trained with a diffusion loss. The paper further conducts a side-by-side evaluation with 22 expert listeners, who preferred the model's output compared to competing models.

**Summary Of The Review:**

As written, I do not think this paper is suitable for ICLR. While the authors claim impressive results on subjective side-by-side evaluations on an impressive number of speech distortions, it is hard to see how others in the ICLR community can build upon this work since there is no new ML technique introduced and the empirical findings are generally not reproducible. At ICLR, there is similar precedent for a well-engineered visual speech recognition system that also did not introduce new ML techniques and whose empirical findings are generally not reproducible (https://openreview.net/forum?id=HJxpDiC5tX). The reviewers recommended that the paper would be more suitable at an applications-oriented venue. I would make the same recommendation in this case.

If the authors would like to revise their work to make it more suitable for ICLR, I would suggest the following:
1) Create a benchmark task for evaluating "universal" speech enhancement. This would involve open-sourcing code for programmatically generating their 55 speech distortions.
2) Propose new automated metrics that holistically evaluates "universal" speech enhancement, and show that they correlate well with subjective side-by-side listener evaluations. This builds upon a long precedent in the speech enhancement literature (PESQ, COVL, STOI, etc) and would be an immensely valuable contribution for the community.
3) Use a simpler and less "hacky" deep learning architecture that is easy for others to replicate by themselves, or release code for it to aid others in their reproducibility efforts.

Finally, I would like to point the authors' attention to several related work in the literature for Table 4. [1], [2], [3], [4] outperforms Universe-Denoise on VoiceBank-Demand. [1], [3] outperforms Universe-Denoise-E on VoiceBank-Demand. [1] outperforms both Universe-Denoise and Universe-Denoise-E on IS20 DNS Challenge. Given that Universe-Denoise-E is an expectation rather than the output of a deterministic model, I think it also helps to provide distribution statistics like median and std to help readers understand its performance. Unfortunately, the related work does cast some doubt on whether a "universal" speech enhancement model can perform as competitively as specialized speech enhancement models, but in my opinion, the presence of better-performing specialized models would not detract from the contribution of a "universal" model.

[1] DPT-FSNet: Dual-Path Transformer Based Full-Band and Sub-Band Fusion Network for Speech Enhancement. Dang et al.

[2] Single-Channel Speech Enhancement using Learnable Loss Mixup. Chang et al.

[3] CMGAN: Conformer-based Metric GAN for Speech Enhancement. Cao et al.

[4] Dual-Branch Attention-In-Attention Transformer for Single-Channel Speech Enhancement. Yu et al.

---

> ### Author Response · Authors · 2022-11-16
> **Reply to comments, part 1**
>
> We thank the reviewer’s comments. We now reply to the reviewer’s concerns grouped by topic:
>
> * Suitability for ICLR – We completely disagree with the reviewer in that papers like the one we submit are not suitable for ICLR. First, we would like to underline several of the novelties introduced by our paper:
>   1. The paper introduces the task of universal speech enhancement and stimulates researchers and practitioners to work in a more unified setting where individual distortions are not treated and evaluated independently with distortion-specific approaches and methodologies. It opens the road for future work on the topic.
>   2. The paper proposes to solve the task with a generative approach. Generative approaches are not the most common approach for distortion-specific tasks and, due to the bias of current objective metrics towards non-generative approaches (see Sec 3.6 in [A]), there is the question whether generative ones can provide good and/or better outputs. We think we convincingly show the latter.
>   3. A number of the techniques we propose in Sec 3.3 are not used previously in the context of speech enhancement or reconstruction, like the use of multiple mixture densities, and some of them are not even used in the context of audio generation itself, like the use of out-of-path losses. We show that they can have an important effect, and think that such effect could transfer to other tasks.
>   4. The proposed approach, to the best of our knowledge, is the first one to use a variance exploding paradigm in the audio domain (only a few in the image domain), and that it probably is the first diffusion approach to produce high-quality speech samples in a task different from vocoding using less than 10 iterations.
>
>   Next, we would like to note that the paper is submitted to the “Applications” track of the conference, and that there are countless submissions accepted for publication in ICLR which do not necessarily introduce core new ML techniques in a generic sense (actually a number of the ICLR works we cite could be regarded as not doing so). We also think this is perhaps a potential bias towards audio-related submissions, as this question of novelty or suitability does not raise so often in other ICLR “Applications” submissions related to computer vision or natural language processing. Overall, we kindly ask the reviewer to reconsider his/her position regarding the novelty of our submission and its suitability for the conference.
>
> * Reproducibility – Unfortunately, we cannot make our data nor code available. We hope this is not an obstacle for acceptance as it has not been in the past with other submissions to ICLR. We make available a large number of enhancement examples (non-cherry-picked) as part of the submission (and will make them publicly-available in the web) in order to facilitate future subjective comparison with our approach (see also Note 1 below).
>
> * “Hacky architecture” – In Appendix B, we have made a tremendous effort to explain all details of the architecture in order to allow a minimally experienced practitioner to replicate it. Nonetheless, if there is some detail that we miss and that could be critical to reproduce our architecture, we would highly appreciate that the reviewer mentions it so that we can add it to the description in Appendix B. Otherwise, we kindly ask the reviewer to reconsider his/her position regarding this topic.
>
> * Create a benchmark and propose new automated metrics – We agree with the reviewer that creating a benchmark and proposing new metrics for universal speech enhancement would be immensely valuable contributions. However, we believe that these achievements are beyond the scope of the current paper (see the summary of contributions above). In general, it is only after a series of papers highlight the importance of a topic that the community sets up proper benchmarks and evaluation pipelines. Prior to that, for universal speech enhancement, there are some issues to be discussed, like core distortions to consider, which publicly-available data sets are more appropriate for each distortion, which metrics to use, or even whether one should just rely on subjective assessment given the complexity of the task. We see our work as the first step towards this more general discussion.

---

> > ### Author Response · Authors · 2022-11-16
> > **Reply to comments, part 2**
> >
> > * UNIVERSE-Denoise-E – We think there is a misunderstanding in what UNIVERSE-Denoise-E does. The reviewer seems to suggest that results of this approach correspond to an expectation taken over multiple calculations of objective scores corresponding to multiple generations per utterance. That is not the case. The results of this approach correspond to calculations of a single objective score per utterance, corresponding to a single final generation per utterance (as computed for baseline approaches and typical speech enhancement works). The crucial point is that, in UNIVERSE-Denoise-E, each single generation corresponds to the expectation (empirical average) of multiple generated signals. Therefore, UNIVERSE-Denoise-E produces a unique final output for every input, not a sample of outputs from which a median or a standard deviation could/should be computed.
> >
> > * Related works – We would like to note that [3] is a concurrent work that appeared on ArXiv in the middle of developing UNIVERSE. Regarding [4], we were not sure if the reported results were computed with narrow-band or wide-band PESQ, and decided to be cautious and not to include them in Table 4 (no details mentioned in [4]). Regarding [1,2], we have now added them to Table 4 and reworked some sentences in the text. We see UNIVERSE is still competitive in some metrics. Nonetheless, we would like to remind the reviewer that these results are just objective scores (which have their own pitfalls, see for instance [A,B]) and correspond only to the task of speech denoising. Importantly, we would like to emphasize that objective metrics are known to seriously penalize generative approaches like UNIVERSE (see [A,B] for more discussion), and that results like the one in Table 4 where UNIVERSE stands out with respect to other generative approaches (top part of Table 4) are hard to obtain and not seen previously.
> >
> > * Universal vs. specialized models – The question of whether a universal model can be competitive with specialized models is precisely what we seek to answer with our subjective test (Table 2). In it, UNIVERSE is directly compared with specialized models in multiple specialized tasks like denoising, declipping, bandwidth extension, etc., and obtains favorable results. In addition, in Table 2, we include the comparison with UNIVERSE-Denoise, which aims to provide insight into the question (Sec 4.2, question 2). The obtained result indicates that, with the same architecture and training protocol, a universal model can be as competitive as a distortion-specialized one. Having said that, we acknowledge that it is always possible that new algorithms tailored to a specific task outperform more generalist ones (we see it as a consequence of the no free lunch Theorem).
> >
> > ---
> >
> > Note 1: We would like to stress that a large variety of audio examples are attached to the current submission, showing the great potential of UNIVERSE to enhance multiple and diverse degraded speech, including real-world distorted speech. Since no reviewer mentions them, we are worried that audio samples, essential for evaluating any generative and/or enhancement algorithm, could have been overlooked.
> >
> > [A] Li & Yamagishi, Noise Tokens: Learning Neural Noise Templates for Environment-Aware Speech Enhancement, INTERSPEECH 2020.
> >
> > [B] Jassim et al., Warp-Q: Quality Prediction for Generative Neural Speech Codecs, ICASSP 2021.

---

> > > ### Comment · Reviewer_XfqC · 2022-11-23
> > > **Thanks for the clarification**
> > >
> > > I had misunderstood how UNIVERSE-Denoise-E works. Thanks to the authors for the clarification, and for revising the paper to elaborate on this point. It's surprising to me how such a significant improvement can be obtained from just taking an expectation over 10 outputs. Is this something particular to the diffusion-based generative approach? For example, if we were to use a Bayesian neural network, or introduce noise in the conventional supervised learning setting, would taking an expectation over the output space also yield significant improvements?

---

> > > > ### Author Response · Authors · 2022-11-28
> > > > **Expectation synthesis of generative models**
> > > >
> > > > Thanks for your interest. Our current thinking is that this expectation strategy could be somewhat applied to other types of generative models like VAEs or GANs, but not to supervised settings. Nonetheless, it is important to note that, despite improving the objective score values, we did not hear a clear improvement when using such strategy. Therefore, we think it is mostly a product of the current evaluation metrics and their negative bias towards generative models. As mentioned in other answers, it has been shown that current objective metrics have a strong tendency to underestimate generative models and overlook plausible speech/audio regeneration, as they are based on exact sample alignment, polarity coherence, etc. (see for example [1,2], especially Sec 3.6 in [1]; one can also see it by comparing generative and non-generative results in Table 4). The intuition is that, here, by taking the expectation, you're making your generative model to become "less generative", and therefore counteracting the underestimation of common objective evaluation measures.
> > > >
> > > > [1] Li & Yamagishi, Noise Tokens: Learning Neural Noise Templates for Environment-Aware Speech Enhancement, INTERSPEECH 2020.
> > > >
> > > > [2] Jassim et al., Warp-Q: Quality Prediction for Generative Neural Speech Codecs, ICASSP 2021.

---

### Official Review · Reviewer_5RnM · 2022-10-24

**Confidence:** 4
**Correctness:** 3
**Technical Novelty And Significance:** 3
**Empirical Novelty And Significance:** 3
**Recommendation:** 6

**Clarity, Quality, Novelty And Reproducibility:**

This paper is well written with high quality results. While diffusion models are not new for speech, this work presents very strong empirical results and universality across distortion types. The main concern would be reproducibility and comparability with past/future work, since the model is only trained on internal data.

**Strength And Weaknesses:**

Strengths
- This paper appears to be the first general model for speech enhancement based on diffusion. Empirical results are very strong. The samples provided in the supplementary materials are impressive especially when tested on the real world samples.
- The presentation of how the authors arrived at their final model architecture and training losses is excellent. The authors motivated the design well. Results of intermediate models are included, serving as a nice reference for future development of audio enhancement models
- It is also informative to include a baseline that is trained with regression loss.

Weaknesses
- The comparison with prior work may not be entirely fair considering that the proposed model is trained on a different dataset. Hence, it cannot be concluded from the paper that UNIVERSE is better than prior work because diffusion is a better objective of enhancement and/or the proposed model architecture is better. In addition, the target UNIVERSE is trained to predict is also enhanced as described in Appendix C, which might be another advantage of the proposed model over baselines. This question can be answered if a) the authors retrain the strongest baseline (e.g.,HiFi-GAN-v2) using the same dataset, or b) train the proposed model using the same public data as the prior work.
- While the authors share the type of distortions used for corrupting clean speech as input, the details of the parameter used in each distortion are not provided (e.g., SNR for colored noise). How clean speech was selected /what noise dataset was used / how many hours of noise are used for training are not specified. It would be hard to reproduce the data setup given the paper.
- The authors mentioned in the appendix that the target was enhanced (Sec C) but results comparing using original speech versus enhanced speech are not provided.


**Summary Of The Paper:**

This work presents a universal diffusion model for speech enhancement for a wide variety of distortions (Table 3 in the appendix), in contrast to prior studies which focused on a limited set of distortion types (cf Table 2). The authors first presented a series of exploration of model architectures and auxiliary losses based on which the authors arrived at their final model. The authors then demonstrated that the proposed model, UNIVERSE, performs better than prior models developed for specific types of distortion as well as ablated models that are trained with regression loss or trained for a specific type of distortion.

**Summary Of The Review:**

It is exciting to see a universal model for speech enhancement based on diffusion with very strong performance. The quality of the work is good. However, it is hard to assess exactly why the proposed model is superior to prior work and how future work can be compared with the proposed model due to the use of non-public datasets.

---

> ### Author Response · Authors · 2022-11-16
> **Reply to comments**
>
> We thank the reviewer’s comments and assessment. In the following, we reply to his/her concerns, grouping them by topic:
>
> * Regarding the comparison with prior work, we want to draw the attention of the reviewer to the fact that “train data does not overlap with the validation partition nor with other data used for evaluation or subjective testing” (Sec 3.1, Data), as this setup puts into great value the results reported here. Since all audios used to assess UNIVERSE performance in Tables 2 and 4 come from different recording conditions, speakers, and noise samples than the ones used for training UNIVERSE, achieving competitive performances is extremely challenging. This is not the case with the considered existing approaches, which are evaluated on matched conditions (in many cases this extends to samples used for subjective testing). We observe that UNIVERSE does convincingly achieve competitive performances in this scenario where there is a mismatch between train and test (for additional pointers on how challenging it is to get good performances in mismatch conditions, see Note A below).
>
> * Regarding the fact of using enhanced targets for training, we want to mention two things. First, that using enhanced targets should be considered a contribution of the proposed approach (and hence it would not be fair to assume that prior approaches include this training strategy). Second, that the fact that we have to use enhanced targets is due to our internal clean data not being of enough quality, which is a characteristic that publicly-available data sets do not present (thus it makes our clean targets more comparable to existing data sets). We do not provide results of original vs. enhanced targets because our original in-house data yielded poor results due to original clean targets still containing some noise and reverb (not studio quality like VCTK). Besides, one should note that some baselines like HiFi-GAN-2 also use enhanced “studio quality” targets as additionally provided in the DAPS data set (Su et al., 2021).
>
> * Regarding distortion details, we use as many hours of noise as speech to produce the data set. SNRs for both real and colored noise are between -5 and 25 dB. We have now added these details in the Appendix. Unfortunately, we cannot make our data nor code available. We hope this is not an obstacle for acceptance as it has not been in the past with other submissions to ICLR.
>
> We thank the reviewer for his/her assessment on clarity, quality, and novelty. Regarding reproducibility, as mentioned, we unfortunately cannot share data nor code. We hope that our answer on the comparison with prior work now highlights the value of the results reported in the paper.
>
> To conclude, we would like to mention that a large variety of audio examples are attached to the current submission, showing the great potential of UNIVERSE to enhance multiple and diverse degraded speech, including real-world distorted speech. Since no reviewer mentions them, we are worried that audio samples, essential for evaluating any generative and/or enhancement algorithm, could have been overlooked.
>
> ---
>
> Note A: An example of mismatch condition can be seen when comparing results, for instance, of HuBERT models on the ASR task using test data that is similar to or different than train data (SUPERB [1] vs. SUPERB-SG [2] benchmarks, respectively). The difference is huge (4% vs. 44% WER). Another comparison point is the mismatch results of a common mask-based speech enhancement pipeline using features that are pre-trained on a different data set [2]. In there, PESQ and STOI values are clearly below what matched systems report, and at a similar level than the ones achieved with FBANK (PESQs around 2.5 and STOIs around 0.93).
>
>
> [1] Yang et al., SUPERB: Speech Processing Universal PERformance Benchmark, INTERSPEECH 2022.
>
> [2] Tsai et al., SUPERB-SG: Enhanced Speech processing Universal PERformance Benchmark for Semantic and Generative Capabilities, ACL 2022.

---

> > ### Comment · Reviewer_5RnM · 2022-12-02
> > **Thanks for the response**
> >
> > I thank the authors for the detailed response
> >
> > > Regarding the comparison with prior work, we want to draw the attention of the reviewer to the fact that “train data does not overlap with the validation partition nor with other data used for evaluation or subjective testing” (Sec 3.1, Data)
> >
> > I agree with the authors that this is indeed significant and I was aware of this in the initial review. Nevertheless, I share the same concerns as reviewer BhNH that comparing with previous work on a fair setting (using the same dataset, not using enhanced targets) is still valuable. While the performance of UNIVERSE is great, we cannot tell whether the gain is brought by having more data, having cleaner target, or having better training objective.
> >
> > > Regarding the fact of using enhanced targets for training, we want to mention two things. First, that using enhanced targets should be considered a contribution of the proposed approach (and hence it would not be fair to assume that prior approaches include this training strategy). Second, that the fact that we have to use enhanced targets is due to our internal clean data not being of enough quality, which is a characteristic that publicly-available data sets do not present (thus it makes our clean targets more comparable to existing data sets).
> >
> > This is yet another reason to include an experiment comparing UNIVERSE with prior work using the public dataset, so the readers can learn better what contributes to the success when everything else is controlled.
> >
> > On the other hand, could the authors share any quantitative evaluation of the quality of the internal clean data and the enhanced version. The authors would be interested in knowing how clean the data have to be to build such a model. For example, is LibriTTS considered not clean enough? Can any reference-free SNR estimator be used to assess the quality before and after enhancement for the targets?
> >
> >
> > Overall, I agree that the paper present a significant improvement over prior works, and I will keep my original rating.

---

### Official Review · Reviewer_yPpC · 2022-10-29

**Confidence:** 3
**Correctness:** 3
**Technical Novelty And Significance:** 3
**Empirical Novelty And Significance:** 3
**Recommendation:** 6

**Clarity, Quality, Novelty And Reproducibility:**

Clarity
- The paper's first parts (abstract and introduction) are well written. However, Section 3 (especially Section 3.2) is not self-consistent within the main document. (it requires reading the appendix to understand the model). Of course, I know we always have page limitations and cannot put all our content together. However, the authors could make more effort by briefly providing the theory and practical parts in the appendix to Section 3.2. Section 3.3 is informative, but we can move some development history parts to the appendix instead.

Quality
- I understand the importance of the subjective evaluation since this is a new setup. However, this paper also has a significant improvement in the modeling part. To validate their model, it is better to have more fair comparisons with the other methods based on the existing database and metrics. Table 4 corresponds to this validation, but due to the difference in the training setup, we cannot compare them. Also, Table 4 is in the appendix and is not considered as the main result.

Novelty
- The prior study does not have such significant variations, and its problem setup is novel.
- The modeling part evolved from the conventional method (described in Section 3.3).

Reproducibility
- The paper has an issue with reproducibility. The database and scoring (based on SESQA, correct me if I'm wrong) are internal. The source code is not available. The source code part is not critical in general; however, this method is very complicated, as described in Section 3.3, and it would not be easy for other people to re-implement this technique.

**Strength And Weaknesses:**

Strength
- the first part (abstract and introduction) of this paper is very well written, and I can easily understand the motivation of this work.
- the related studies for diffusion-based models are adequate.
- although the proposed architecture and training method are complicated, the authors provide detailed explanations of how they evolved their architecture and method with the experimental evidence.
- task setup and their modeling are novel

Weaknesses
- The paper has weak reproducibility (see below).
- The explanation of the model (especially Section 3.2) and some experimental discussions (e.g., Table 4) are not self-consistent (see my comments below).
- The modeling efforts and task setup should be separately evaluated. For example, it is better to show the effectiveness of the proposed model with the same training data for the VCTK+demand experiments since the training data is different. We could also train the conventional models with the same training data.
- another significant distortion comes from interference speakers. I expect the paper to have bubble noise, but speech separation should also be considered if we call this method "universal" speech enhancement.



**Summary Of The Paper:**

This paper focuses on various types of speech distortions (a total of 55) and aims to realize universal speech enhancement based on score-diffusion-based models. The paper carefully describes their complicated architecture based on the various attempts motivated by related studies and incrementally improves the architecture described in Section 3. Overall, their proposed neural network architecture and training methods are novel. The paper also shows the effectiveness of the proposed method by comparing various other state-of-the-art methods with their speech distortion setups based on the subjective preference test.

**Summary Of The Review:**

The paper has sufficient novelties in the problem setup and modeling parts. However, I'm concerned about the lack of reproducibility of this method and the fair comparisons with the existing methods in the conventional metrics.

Other suggestions:
- It is better to discuss speech separation since it is regarded as another critical speech distortion in the broad sense.
- Section 2, the last paragraph: I'm not very sure about the effectiveness of the generative model method for this setup. Since we could simulate most of these distortions, we can make a discriminate model with the artificially created pair data. Please clarify this point.
- Why does the evaluation metric not include PESQ?
- Figure 1(a): I'm very curious how we predict "g" since $\log (\sigma)$ seems to be a scale, and it may not have enough information to predict it. Probably, I misunderstand $\log (\sigma)$. It would be great if the authors clarify it.

---

> ### Author Response · Authors · 2022-11-16
> **Reply to comments, main text**
>
> We thank the reviewer’s comments and feedback. We now reply to both major and minor questions.
>
> Major questions:
>
> * Clarity – We thank the reviewer for the suggestion of moving swapping parts between the Appendix and Sec 3. This was something we actually considered, but we finally decided that it made more sense to place the general theory of score-based diffusion and the known instantiation of variance explosion in the Appendix (but leaving a summary in Sec 3.2), and spend a larger part of the main paper in motivating and explaining which architectural and conceptual changes we introduce to bring the approach to the next level (Sec 3.3). Nonetheless, if deemed necessary, we can rewrite those parts for the final iteration of the paper.
>
> * Quality – We would like to note that Table 4 corresponds to only a part of the validation (speech denoising), while the subjective evaluation embraces a more universal case with other distortions. Regarding objective metrics, we do not find them reliable to measure perceptual quality of enhanced speech. First, it is well-known that objective metrics like PESQ and STOI have their limitations for assessing denoising or dereverberation, respectively. Second, it has been shown that current objective metrics have a strong tendency to underestimate generative models and overlook plausible speech/audio regeneration, as they are based on exact sample alignment, polarity coherence, etc. (see for example [1,2], especially Sec 3.6 in [1]; one can also see that by comparing generative and non-generative results in Table 4). Third, there are no well-established metrics nor data sets for evaluating degradations beyond background noise or reverb, and such degradations are precisely a core part of the task we address. For these and other reasons, we decide to perform a subjective test as the main form of assessing enhancement quality. Regarding fair comparisons, we want to draw the attention of the reviewer to the fact that “train data does not overlap with the validation partition nor with other data used for evaluation or subjective testing” (Sec 3.1, Data), as this setup puts into great value the results reported here. Since all audios used to assess UNIVERSE performance in Tables 2 and 4 come from different recording conditions, speakers, and noise samples than the ones used for training UNIVERSE, achieving competitive performances is extremely challenging. This is not the case with the considered existing approaches, which are evaluated on matched conditions (in many cases this extends to samples used for subjective testing). For additional pointers on how challenging it is to get good performances in mismatch conditions, see Note A below.
>
> * Novelty – We thank the reviewer for his/her assessment on novelty, especially regarding problem setup and modeling. As a side note, we would like to highlight that our approach, to the best of our knowledge, is the first one to use a variance exploding paradigm in the audio domain (only a few in the image domain), and that it probably is the first diffusion approach to produce high-quality samples in a task different from vocoding using less than 10 iterations.
>
> * Reproducibility – It is true that we use an internal measure like SESQA for scoring, but we complement that with three additional well-known and publicly-available measures: COVL (which includes PESQ), STOI, and WARP-Q. Our thinking around objective measures is mentioned in the paper and in the point regarding “Quality” above, and we essentially use those only as guidelines for model development. Regarding the other aspects, unfortunately, we cannot make our data nor code available. We hope this is not an obstacle for acceptance as it has not been in the past with other submissions to ICLR.
>
> Minor questions:
>
> * We are unsure to which speech separation works the reviewer is referring to. We can discuss that topic in the paper and add necessary references when those are clear.
>
> * The observation is based on the tendency we see on tasks like bandwidth extension, silent gap regeneration, or codec enhancement, where generative approaches typically outperform non-generative ones. The cited works of Polyak et al. (2021), Pascual et al. (2019) and Zhang et al. (2021b) provide some good discussion.
>
> * We believe the reviewer is referring to Table 2. In it, we decided to exclude PESQ, as COVL already includes it and is correlated to it (Loizou 2013).
>
> * $\textbf{g}$ is the output of SigmaBlock, which is a multi-layer perceptron on top of random Fourier features (Appendix B). This is the same strategy as used by Rouard & Hadjeres (2021).

---

> > ### Author Response · Authors · 2022-11-16
> > **Reply to comments, notes and references**
> >
> > We would like to highlight that a large variety of audio examples are attached to the current submission, showing the great potential of UNIVERSE to enhance multiple and diverse degraded speech. Since no reviewer mentions them, we are worried that audio samples, essential for evaluating any generative and/or enhancement algorithm, could have been overlooked.
> >
> > Note A: An example of mismatch condition can be seen when comparing results, for instance, of HuBERT models on the ASR task using test data that is similar to or different than train data (SUPERB [3] vs. SUPERB-SG [3] benchmarks, respectively). The difference is huge (4% vs. 44% WER). Another comparison point is the mismatch results of a common mask-based speech enhancement pipeline using features that are pre-trained on a different data set [4]. In there, PESQ and STOI values are clearly below what matched systems report, and at a similar level than the ones achieved with FBANK (PESQs around 2.5 and STOIs around 0.93).
> >
> > [1] Li & Yamagishi, Noise Tokens: Learning Neural Noise Templates for Environment-Aware Speech Enhancement, INTERSPEECH 2020.
> >
> > [2] Jassim et al., Warp-Q: Quality Prediction for Generative Neural Speech Codecs, ICASSP 2021.

---

> > > ### Comment · Reviewer_yPpC · 2022-11-29
> > > **Thanks for your response**
> > >
> > > Thanks for your response.
> > > I particularly agree with the authors' points on the issue of the objective evaluation with the generative model and the generalization capability of the generative model under mismatched conditions.
> > >
> > > About speech separation, to make my comments clear, I just want to use the word, speaker separation (multi-talker separation) in https://ieeexplore.ieee.org/abstract/document/8369155/. I'm not referring to some specific papers on speaker separation, but I'm curious about the general discussion about an extension (or an existing ability) of dealing with interfering speakers based on this method.

---

> > > > ### Author Response · Authors · 2022-12-01
> > > > **Thanks for the clarification**
> > > >
> > > > Thanks for your feedback and the clarification on speaker separation. We agree that speaker separation is an important task that requires further research. Nonetheless, we see it as a complementary task to speech enhancement (and for instance have some doubts of whether a diffusion approach would be the most suitable for separating speakers). Regarding how UNIVERSE performs when presented with utterances with overlapping speakers, we can comment on some empirical findings. Although speaker separation is not our target, we did include some speech (e.g., cafeteria chatter) as background noise when constructing our data set. As a result, when the level difference between two or more concurrent speakers is high, we observe that UNIVERSE suppresses the ones with less energy. However, on the other hand, when the levels are comparable, UNIVERSE treats concurrent speech as a single signal to be enhanced, and therefore tries to remove other distortions while improving the high-energy speech sources. The result in that case is generally good (according to informal listening), but presents room for improvement in some specific cases. In particular, we observe a few cases where UNIVERSE oscillates between suppression and enhancement mode for concurrent speech, which sounds unnatural.

---

### Official Review · Reviewer_BhNH · 2022-10-31

**Confidence:** 4
**Correctness:** 2
**Technical Novelty And Significance:** 3
**Empirical Novelty And Significance:** 2
**Recommendation:** 6

**Clarity, Quality, Novelty And Reproducibility:**

The paper is largely clear and the writing quality is good. The novelty is limited to some extent. I did not see any comments on reproducibility like code, model release etc.


**Strength And Weaknesses:**

Strengths


1. Diffusion based approaches have been very successful in generative tasks in other domains. While their applications to the audio domain are starting to show up, the success is perhaps not yet at the level in other domains like images. To this end, the authors' attempt on using diffusion based learning for speech enhancement is a good idea. Extending it to degradations beyond additive noise also makes sense.


2. I liked that authors did subjective tests as well.


Weaknesses


1. The experimental section leaves quite a few things unanswered. More on it in the review summary below - for which I would like to see the author's response.


2. I think the novelty of the diffusion-learning process is limited. It largely follows prior works.


3. Some questions below are part of the weaknesses.


**Summary Of The Paper:**

The paper proposes diffusion models based speech enhancement. In particular, it extends enhancement to a large number of degradations beyond additive noise. The degradation includes common speech degradations such as codec artifacts, bandwidth reduction, reverb etc. The proposed approach puts together different pieces of diffusion-based learning in the context of speech enhancement and then different variations on the top of the base approach are applied to achieve improved performances.

**Summary Of The Review:**

This work proposes speech enhancement through diffusion based approaches. It trains a single model to learn from a large number of speech degradations.


1. There is a waveform direct waveform based loss too - the significance of which seems to be very high. Doesn’t that default the whole process to so many other speech enhancement models where MSE  on waveform or spectrograms is common. Moreover, there are other latent targets which are also added like pitch, VAD, loudness. One of the versions uses noise level information too. I wonder how much some of the other prior methods can be improved just by adding these additional latent targets and  these auxiliary information.



2. In Section 4.2.1 authors discuss a regression based approach with Universe architecture. Along with waveform and STFT losses are other losses  (as in UNIVERSE like latent targets etc, ) used here ?


3. I am not sure how fair the comparison is with prior works here.
First of all the proposed model is likely orders of magnitude bigger than several of the prior works.
In fact, some of them might be causal as well (which proposed model is not) and causality can have significant impact on performance.
Moreover, the training data is different for different algorithms whereas here a large corpus of private data is used. But importantly, several of these prior works were never trained on such degradations.  I wonder what performance would look like if we just take them and train them with all these other degradations.
I think decoupling the point above is necessary because it will tell us whether variety in speech degradations is important or diffusion-based learning is bringing something to the table.



3. It would be good to show how this model performs in really difficult conditions. For example, a large number of state of the art methods do a really good job on enhancement but struggle in low-SNRs.  A model like the current one (very large, non-causal) are clearly not the most practical speech enhancement systems which often have real-time uses. Nevertheless, this work puts together a whole bunch of pieces to build the enhancement system. Considering this, stronger results would have been expected, and especially in situations where current state of the art struggles. How does this model compare against prior works in really difficult quality conditions?



4. While I like subjective tests in the paper, I think adding MOS subjective scores would add a lot of value. It will clarify a lot of different aspects. Comparative tests do not give an idea of absolute performance of UNIVERSE or others.

-- updates after rebuttal --
updated the score and comments.

---

> ### Author Response · Authors · 2022-11-16
> **Reply to comments, main text**
>
> We thank the reviewer’s comments, and reply to them following the same numbering as above:
>
> 1. We claim that our approach including a waveform-based loss or additional targets does not default to the same as in other speech enhancement models. To test this claim, we consider UNIVERSE-Regress, which corresponds to the exact same model as UNIVERSE but replaces the generative process (diffusion-based) by a common waveform loss (MSE+MRSTFT). This set of experiments is explained in Sec 4.2.1, and the result in Table 2 shows that employing the generative process is important and allows to go further than more common models with the same auxiliary targets. Furthermore, we want to note that both the use of auxiliary losses and how they are specifically employed (placed out of the main signal processing path) are important contributions of our approach (Table 1 and Sec 3.3, experiments S5 and A3).
>
> 2. Yes, the model is exactly the same as UNIVERSE (that is, including out-of-path losses in Fig 1B) but replaces the diffusion approach by the best regression approach we could find (that is, waveform and STFT losses in Fig 1A). We did an extensive search for different combinations of common architecture, loss, and hyperparameter choices, and found only equivalent or worse results.
>
> 3. Understanding the importance of a variety of speech degradations is the principal motivation behind considering UNIVERSE-Denoise, which corresponds to the exact same model as UNIVERSE but trained and evaluated only on the denoising task. This set of experiments is explained in Sec 4.2.2, and the result in Table 2 shows that, although the multiple-distortion model tends to be preferred over the single-distortion one on the denoising task, that preference is not statistically significant. Furthermore, we want to draw the attention of the reviewer to the fact that “train data does not overlap with the validation partition nor with other data used for evaluation or subjective testing” (Sec 3.1, Data), as this setup puts into great value the results reported here. Since all audios used to assess UNIVERSE performance in Tables 2 and 4 come from different recording conditions, speakers, and noise samples than the ones used for training UNIVERSE, achieving competitive performances is extremely challenging. This is not the case with the considered existing approaches, which are evaluated on matched conditions (in many cases this extends to samples used for subjective testing). We observe that UNIVERSE does convincingly achieve competitive performances in this mismatch scenario (for additional pointers on how challenging it is to get good performances in mismatch conditions, see Note A below).
>
> 4. Our subjective test set consists of randomly chosen examples among all the available ones in the demo web pages of the existing approaches. Therefore, in the subjective test, the range of distortion levels is the same for both UNIVERSE and every competing approach. We consider that the samples used for the subjective test comprise a good range of SNRs and particularly include low-SNR cases. In the ZIP file attached to the submission (see Note B) we also provide several validation samples (non-cherry-picked) using public data sets that feature low SNRs. Finally, we disagree with the reviewer in that non-causal, large models are not the most practical nor that speech enhancement primarily has real-time uses. We believe there are plenty of cloud- or plugin-based applications for both amateur and professional editors where non-causal, large models are practical and can excel. We would also like to remind that UNIVERSE can run above 10 times faster than real time without much loss in quality given appropriate hardware (Fig 3 and Sec 4.3).
>
> 5. It is true that comparative tests do not give an idea of absolute performance. However, we think that they are the best possible option to show how one model (UNIVERSE in this case) compares with the others. Comparative tests have also many known and well-documented advantages with respect to MOS tests, including easiness for raters to perform binary decisions instead of more fine-grained ones, forcing a choice in difficult cases, or clear methods to calculate statistical significance. Also, note that performing a MOS test with different materials (demo websites of the compared models feature different examples than each other) and without a common reference (real-world distorted audios do not have a target reference) would have been misleading and not informative.
>
> Finally, as a short comment on the novelty of the diffusion-learning process, we would like to highlight that our approach, to the best of our knowledge, is the first one to use a variance exploding paradigm in the audio domain (only a few in the image domain), and that it probably is the first diffusion approach to produce high-quality samples in a task different from vocoding using less than 10 iterations.

---

> > ### Author Response · Authors · 2022-11-16
> > **Reply to comments, notes and references**
> >
> > Note A: An example of mismatch condition can be seen when comparing results, for instance, of HuBERT models on the ASR task using test data that is similar to or different than train data (SUPERB [1] vs. SUPERB-SG [2] benchmarks, respectively). The difference is huge (4% vs. 44% WER). Another comparison point is the mismatch results of a common mask-based speech enhancement pipeline using features that are pre-trained on a different data set [2]. In there, PESQ and STOI values are clearly below what matched systems report, and at a similar level than the ones achieved with FBANK (PESQs around 2.5 and STOIs around 0.93).
> >
> > Note B: We would like to stress that a large variety of audio examples are attached to the current submission, showing the great potential of UNIVERSE to enhance multiple and diverse degraded speech, including real-world distorted speech. Since no reviewer mentions them, we are worried that audio samples, essential for evaluating any generative and/or enhancement algorithm, could have been overlooked.
> >
> >
> > [1] Yang et al., SUPERB: Speech Processing Universal PERformance Benchmark, INTERSPEECH 2022.
> >
> > [2] Tsai et al., SUPERB-SG: Enhanced Speech processing Universal PERformance Benchmark for Semantic and Generative Capabilities, ACL 2022.

---

> > > ### Comment · Reviewer_BhNH · 2022-11-29
> > > **Thanks for the response**
> > >
> > > I have increased my score to reflect the clarifications given in the rebuttal. Some concerns remain including those brought by other reviewers - around experimental results, reproducibility, disentanglement of universal speech enhancement with respect to the architecture and data etc.
> > > Also, I would like to state that the paper is indeed a suitable ICLR submission (this assessment came in one of the reviews). While one can argue for its acceptance/rejection based on its merits/concerns, but otherwise I believe this paper is suitable for ICLR.

---

### Decision · Program_Chairs · 2023-01-20

**Decision:**

Reject

**Justification For Why Not Higher Score:**

Reproducibility and fair comparison were identified as significant enough to recommend rejection.

**Justification For Why Not Lower Score:**

N/A

**Metareview: Summary, Strengths And Weaknesses:**

The authors present a model for ‘universal’ speech enhancement, to address a wide-range of distortions at the same time. The model itself is a diffusion-based denoiser that uses score-based diffusion.

The reviewers agree that this is one of the first papers that addresses all the varied distortions in a single model, with good empirical results. While the term ‘universal’ may need reconsideration since the work does not address multitalker separation, which is a very popular use case for speech frontends.

There is also some novelty in the application of score-based diffusion models to enhancement, and the overall model architecture, which is presented in a nice way despite being complex. Although score based diffusion is from prior work, reviewers agreed that its application to enhancement is somewhat novel.

One of the shortcomings raised by multiple reviewers is that it lacks enough comparisons. For instance, there are various components to the model formulation, architecture, losses, and the data. It is not always clear what parts are contributing to the final performance. The reviewers noted that the mismatch in training data and the differences in model capacity likely makes the presented comparisons with prior art unfair. While the authors argue in their rebuttal that their model is evaluated on held-out data, that does not fully address the fact that the models compared against are at a disadvantage. This makes it difficult to assess what parts of the model are contributing to the gains (discounting the universal nature of the new model, which is indeed unique to the presented work).

Along similar lines, the authors do not train on public datasets nor make the code available. The presented technique is complex, all of which makes it a bit challenging to replicate. What would have helped is if the authors presented results also using public datasets for training, to make it easier to compare, and would have served as a useful target for researchers trying to replicate the technique. This could have also made comparisons with prior art more fair.

**Summary Of Ac-Reviewer Meeting:**

The main discussion points during the meeting:
- Novel task setup, which is the main novelty of the presented work. Diffusion based approach has been applied for SE, but not the exact formulation that the authors use. There is also some novelty in model architecture.
- The authors claim that their approach allows the model to converge in 4-8 steps. But there are existing techniques to speed up convergence of diffusion based models for enhancement.
- Reproducibility is really challenging. It is unclear how researchers can compare with this technique going forward. Although, universal model is interesting, it is hard to reproduce given the complexity.
- Experimental validation is not great, and neither do the authors address it head on in their rebuttal. They also don’t train their model on publicly available datasets.
- The presented results are not fair because they use more data and a different architecture. So it is unclear if the gains are from the universal task, the diffusion model, or the overall architecture.